# ATF3 acts as a rheostat to control JNK signalling during intestinal regeneration

Jun Zhou[1], Bruce A. Edgar[2] & Michael Boutros[1]

Epithelial barrier function is maintained by coordination of cell proliferation and cell loss, whereas barrier dysfunction can lead to disease and organismal death. JNK signalling is a conserved stress signalling pathway activated by bacterial infection and tissue damage, often leading to apoptotic cell death and compensatory cell proliferation. Here we show that the stress inducible transcription factor ATF3 restricts JNK activity in the *Drosophila* midgut. ATF3 regulates JNK-dependent apoptosis and regeneration through the transcriptional regulation of the JNK antagonist, Raw. Enterocyte-specific ATF3 inactivation increases JNK activity and sensitivity to infection, a phenotype that can be rescued by Raw over-expression or JNK suppression. ATF3 depletion enhances intestinal regeneration triggered by infection, but does not compensate for the loss of enterocytes and ATF3-depleted flies succumb to infection due to intestinal barrier dysfunction. In sum, we provide a mechanism to explain how an ATF3-Raw module controls JNK signalling to maintain normal intestinal barrier function during acute infection.

[1] German Cancer Research Center (DKFZ), Division Signaling and Functional Genomics and Heidelberg University, Department for Cell and Molecular Biology, Medical Faculty Mannheim, Im Neuenheimer Feld 580, 69120 Heidelberg, Germany. [2] German Cancer Research Center (DKFZ)-Center for Molecular Biology Heidelberg (ZMBH) Alliance, 69120 Heidelberg, Germany. Correspondence and requests for materials should be addressed to M.B. (email: m.boutros@dkfz.de).

ntestinal epithelia in most animals undergo constant renewal under normal homeostatic conditions[1]. This turnover rate increases upon infection and damage to the gut epithelium[2,3]. Turnover of the gut epithelium is dependent on intestinal stem cells (ISC), which can differentiate into nearly all intestinal cell types[4]. Cellular programmes such as proliferation, differentiation and apoptosis interplay in the regenerating intestinal epithelium[5]. Environmental stresses, such as oxidative stress and microbial infection from food-borne pathogens, can cause intestinal inflammation and systemic stress responses, factors that are also associated with premature ageing and age-related diseases[6–8]. Enteric infection induces an increase in the stem cell division rate to replenish damaged tissue[9,10]. Infection also triggers stress response pathways, which can lead to enterocytes loss through delamination and apoptosis[11]. Dysregulation of stress, immune and inflammatory signalling, and stem cell proliferation result in impaired epithelial renewal and subsequent barrier dysfunction, which increases animal mortality. However, it is mostly unknown how apoptotic signalling and stem cell activity (for example, after infection with pathogens) is buffered to prevent barrier dysfunction of the epithelium.

*Drosophila melanogaster* is an important model system to dissect the integration of intestinal regeneration, immunity and stress response, which are crucial processes for tissue homeostasis, animal health and ageing[7,8,12]. Innate immune responses in *Drosophila* trigger several pathways, including JNK, Imd, Toll and JAK/STAT signalling that can induce expression of antimicrobial peptides and other immune defense programmes[13–16]. Infection by the pathogenic bacterium *Pseudomonas entomophila* induces local antimicrobial peptide expression in the intestinal epithelium and a systemic immune response through the fat body[17]. *P. entomophila* can cause severe damage to the *Drosophila* intestine and result in epithelial dysfunction that impairs immune and repair programmes, and eventually organismal death[18]. Previous studies in the *Drosophila* midgut showed that ISCs proliferate rapidly to produce new entoerocytes for epithelial renewal, and that homeostasis is maintained by activating JNK, EGFR, JAK/STAT and BMP signalling pathways, which are activated in response to oxidative stress, bacterial infection and ingestion of toxins such as dextran sodium sulfate or bleomycin[9,19–29]. JNK influences *Drosophila* gut regeneration by promoting stem cell proliferation[25,30], and activated JNK can also cause apoptosis of enterocytes[11], indicating that JNK signalling has a complex function in tissue homeostasis. The *puckered* (*puc*) gene is a target of JNK signalling that encodes a JNK phosphatase and thereby mediates a negative feedback loop regulating JNK activity[31]. Nevertheless, how JNK activity is triggered and then controlled in the intestinal epithelium to coordinate enterocyte death and intestinal regeneration is not well understood. The conserved transcription factor, ATF3 is induced by a variety of stress signals including cytokines, genotoxic agents or physiological stress[32,33], and is important in metabolic and immune homeostasis in *Drosophila*'s gut epithelium[34,35]. A previous study reported increased JNK activity in *Drosophila* atf3[76] mutant larval guts[35], but beyond this little is known about the molecular mechanisms linking JNK to ATF3 or ATF3 function in gut homeostasis.

Here we investigate the role of ATF3 in the control of intestinal JNK activity. We find that ATF3 controls JNK activity through direct transcriptional regulation of a JNK antagonist, Raw. ATF3 and Raw function together to restrain JNK mediated enterocyte apoptosis and tissue regeneration. Interestingly, flies that over-express ATF3 or Raw survive better than wild-type flies after infection with *P. entomophila*. Conversely, flies deficient in either gene are more susceptible to infection as a consequence of loss of epithelial cells and barrier dysfunction. The infection suscept-ibility of ATF3-deficient flies can be rescued by forced expression

of *Raw* or dominant negative JNK, indicating that JNK hyper-activation is the main dysfunction in these flies. Thus, our study uncovers an essential autonomous role of ATF3-Raw-JNK signalling in controlling cell survival and stress responses in intestinal enterocytes.

## Results

**ATF3 is a stress response gene in *Drosophila* intestine**. In a screen for mediators of intestinal homeostasis, we identified ATF3 as a strong modulator of ISC proliferation. In humans, ATF3 has been described as a stress sensor for a wide range of insults, including genotoxic stress, ER stress and inflammatory reactions[32,36–38]. In *Drosophila*, ATF3 is expressed in multiple tissues of the adult and is particularly highly expressed in the digestive tract including the midgut, hindgut and crop (Supplementary Fig. 1a).

To identify which cells in the midgut express ATF3, we utilized an *ATF3::GFP* transgenic line[35] that harbours an ATF3-EGFP fusion protein under the control of the genomic ATF3 regulatory sequence (Atf3[gBAC]). We found that enterocytes with large-nuclei showed a strong GFP-signal, while Delta-positive stem cells (ISCs) were only weakly GFP-positive. In contrast, GFP was not detected in Prospero-positive enteroendocrine (EE) cells (Fig. 1a,b). The ATF3::GFP signal was greatly reduced by RNAi against ATF3 (Fig. 1c,d), indicating that it was specific. These results indicate that ATF3 is mainly expressed in enterocytes, and to a weaker extent in ISCs.

Next, we used RT-qPCR (quantitative PCR with reverse transcription) to determine whether ATF3 is induced in the *Drosophila* midgut by various stresses, including infection with the Gram-negative bacteria *Pseudomonas entomophila*, ingested paraquat (induces oxidative stress) and ageing. We observed that *ATF3* mRNA is induced several fold following infection with *Pseudomonas entomophila* (*P.e.*), or by paraquat ingestion (Fig. 1e,f). In addition, *ATF3* mRNA levels gradually increased during ageing (Supplementary Fig. 1b). Consistent with this, we found that ATF3 is increased in ageing intestine in a published transcriptome analysis[39]. These data indicate that ATF3 is a stress inducible gene in the *Drosophila* intestine.

**ATF3 in enterocytes restricts ISC proliferation**. Since ATF3 influences stem cell proliferation and is strongly expressed in enterocytes, we further investigated its role in the regulation of ISC proliferation using cell-type specific loss-of function analysis. After depleting ATF3 using two independent ATF3 RNAi constructs under the control of the inducible, enterocytes-specific Gal4 driver *MyoIA[ts,25]*, we observed a significant increase in the number of gut mitoses, as assayed using anti-phospho-histone H3 (Fig. 1g-i, and Supplementary Fig. 1c,d). This increase in mitoses in ATF3-depleted midguts increased with time over a 15-day period, suggesting a progressive breakdown of homeostasis (Supplementary Fig. 1e). In addition, we observed no significant change in ISC division in the intestine of EE or VM specific *ATF3* RNAi flies (Supplementary Fig. 1i–n).

The *atf3[76]* mutant allele lacks ATF3's bZIP domain and homozygous mutant females die during larval stages[35]. A small fraction of *atf3[76]* hemizygous mutant males survive to adulthood but show a severe reduction in body size[35]. Similarly to RNAi, a significant increase in mitotic cell number was observed in the midguts of *atf3[76]* hemizygous mutants (Fig. 1j–l). Moreover, we observed an accumulation of cells expressing *esg-lacZ*, a marker of ISCs and EBs (Supplementary Fig. 1f–h). However, we could not exclude the possibility that gut homeostasis defects in the viable but sickly *atf3[76]* hemizygous mutants were due to

developmental defects, and therefore in further analysis we relied more on conditional tissue-specific depletion of ATF3 by RNAi. Hence, tissue-specific knockdown experiments support a model whereby ATF3 functions in the fly gut to restrict ISC proliferation.

**ATF3 depletion-induced ISC mitosis is JNK dependent.** To identify signalling pathways required for ATF3-mediated proliferation of ISCs, we assayed the expression of different pathway components and their transcriptional targets. We observed that mRNAs encoding the JNK-pathway components *Kayak* (*kay; D-fos*) and *Puckered* (*puc; Jun kinase phosphatase*)

were induced upon ATF3 depletion in enterocytes, as well as the EGFR ligands *Vein* and *Spitz* and the JAK/STAT ligand *Upd3* (Supplementary Fig. 2a). Consistently, the downstream JAK/STAT target *Socs36E* was highly upregulated (Supplementary Fig. 2a). We also observed upregulation of *Hh*. No effects on the *Dpp* signalling target Daughters against dpp, (Dad) were observed, but the transcription factor (Mothers against dpp, Mad) was downregulated (Supplementary Fig. 2a). Consistent with the increase in *kay* and *puc* expression, we observed a strong induction of the *puc* reporter gene, *puc-lacZ* (ref. 40), in ATF3-depleted enterocytes (Fig. 2a,b). Increased *puc-lacZ* expression was also found in *atf3*[76] mutant males (Fig. 2c,d). These results are consistent with a previous report of increased *puc-lacZ*

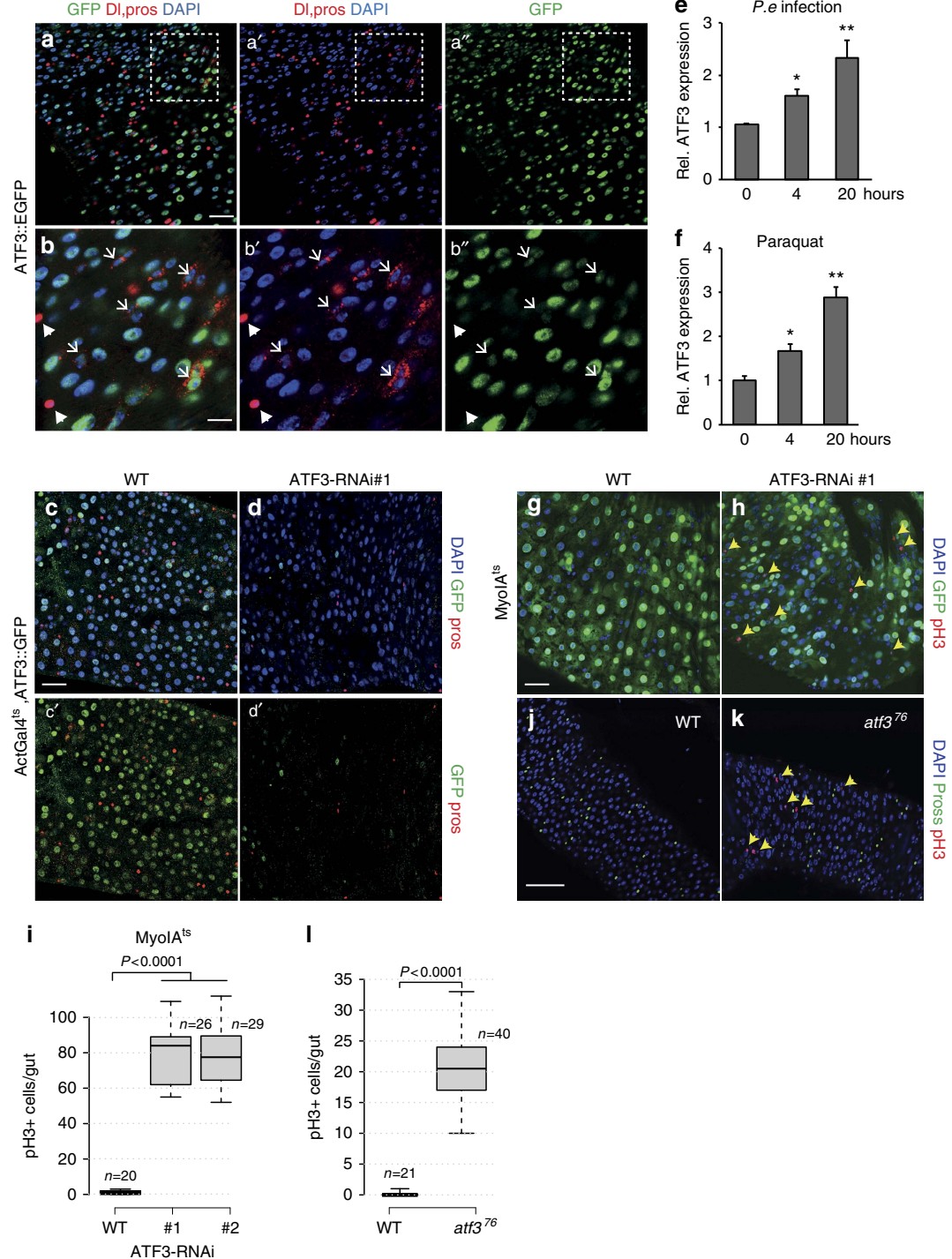

expression in *atf3*[76] mutant larvae gut[35]. Similarly, we observed elevated phospho-JNK in enterocytes upon ATF3 depletion (Supplementary Fig. 2c,d). These results indicate that ATF3 depletion triggers JNK and JAK/STAT signalling in the fly's midgut.

We next tested whether ATF3 depletion leads to JNK activation in enterocytes in a cell autonomous or non-cell autonomous manner using the clonally inducible *esg* flip-out system (*esg F/O*) (ref. 10). With this induction system *esg* positive cells and all of their newborn progeny will express Gal4, UAS-GFP and *ATF3* RNAi after a temperature shift, whereas previously existing enterocytes will be GFP- and RNAi-negative. We first performed phospho-JNK staining and found JNK signalling was only activated in *ATF3* RNAi clones, but not in neighbouring wild-type (control) cells (Fig. 2e and Supplementary Fig. 2e). Similarly, we observed an induction of *puc-lacZ* in ATF3 RNAi expressing clones (Supplementary Fig. 2b). This indicates that ATF3 acts cell autonomously to restrain JNK activity in enterocytes.

We then tested the epistatic relationship between ATF3 and JNK signalling by simultaneous depletion of ATF3 and suppression of JNK pathway components (*UAS-Kay-RNAi* [Kayak RNAi] or *UAS-Bsk*[DN] [dominant negative *Basket*]) in enterocytes. Suppressing JNK signalling in conjunction with expressing *ATF3 RNAi* effectively suppressed the hyperproliferation of stem cells caused by *ATF3 RNAi* alone (Fig. 2f). To further confirm that sustained JNK activation contributes to the mitoses driven by ATF3 depletion, we fed ATF3-depleted flies the JNK inhibitor SP600125. This also strongly suppressed the increase in stem cell mitosis in the ATF3 depleted midguts (Fig. 2g). Taken together, these results indicate that sustained activation of JNK is required for the ISC overproliferation that occurs in ATF3-depleted midguts.

**STAT contributes to ATF3 loss induced ISC division.** Upon JNK activation in the fly's midgut, the Upd2 and Upd3 cytokines are expressed and trigger a non-cell autonomous increase in ISC division[9,10,28]. Indeed, Upd2,3 and JAK/STAT signalling are required for midgut homeostasis and damage-dependent regeneration[9,10]. Given that ATF3 antagonizes JNK to control ISC proliferation, we investigated whether ISC proliferation after loss of ATF3 is dependent on JAK/STAT signalling. To confirm the upregulation of Upd3 in enterocytes, we depleted ATF3 using RNAi and the *MyoIA*[ts] driver, and monitored an *Upd3-lacZ* reporter gene[10]. As shown in Fig. 2h,i, we observed very high levels of *Upd3-lacZ* expression in *MyoIA-GFP* positive enterocytes of ATF3 depleted midguts. In addition, elevated

*Upd3-lacZ* levels were observed in ATF3 RNAi-expressing clones, but not in adjacent control cells (Fig. 2j and Supplementary Fig. 2f). This indicates that ATF3 cell-autonomously restrains Upd3 expression in enterocytes.

Next, we monitored JAK/STAT activity in ATF3-depleted midguts using a transgenic reporter, *10X-STAT-GFP* (ref. 41). Expression of *ATF3-RNAi* in the midgut for either 1 or 5 days resulted in a strong induction of STAT-GFP expression (Fig. 2k,l and Supplementary Fig. 2g–k). To test the functional relevance of the observed Upd3 induction, we combined an *Upd3-RNAi* transgene with *ATF3 RNAi* and examined the mitotic index. We observed that co-depletion of Upd3 effectively suppressed the proliferation of ISCs caused by ATF3 depletion (Fig. 2m). These data support a model in which ATF3 depletion increases JNK activity, which in turn triggers Upd3 production to stimulate ISC activation and division.

**The *Raw* gene is a direct transcriptional target of ATF3.** To understand how ATF3 regulates JNK signalling, we performed a chromatin immunoprecipitation (ChIP-seq) experiment using midguts from ATF3::EGFP BAC transgenic flies (data not shown). From the analysis for potential binding sites, we observed a binding peak in the promoter region of *ATF3* (Supplementary Fig. 3a). The bound region at the *ATF3* promoter, but not at its coding region, was validated by chromatin immunoprecipitation qPCR (ChIP-qPCR) (Supplementary Fig. 3b). These results indicate ATF3 is able to regulate its own expression. In addition, we found an ATF3 binding peak in the first intron of the *Raw* locus (Fig. 3a). Raw, which only shows limited homology to genes in vertebrate species, has been previously implicated as an antagonist of JNK signalling[42,43]. ChIP-qPCR experiments confirmed the binding of ATF3 at the first intron of the *Raw* locus. In contrast, no significant binding was observed in the coding region of the first exon or other un-related genomic regions (Fig. 3b). To assess the regulation of *Raw* expression by ATF3, we examined its expression after bacterial infection (*P.e*) or ageing in enterocytes of *ATF3-RNAi* or *ATF3*-overexpressing flies. In unchallenged conditions, we observed a small reduction of *Raw* expression in the ATF3 depleted midguts (Fig. 3c). Interestingly, *Raw* expression was induced in response to infection (Fig. 3c) and this induction could be reduced by ATF3 depletion (Fig. 3c). Conversely, overexpression of ATF3 strongly induced *Raw* expression (Fig. 3c). Similar effects were observed in the guts of ATF3 depleted or overexpression flies during ageing (Supplementary Fig. 3c). We noticed that *Raw* expression was slightly decreased in the intestines of immune deficient flies (*PGRP-LC*[7457]) at 0 and 4 h post

**Figure 1 | ATF3 expression is induced in the intestine upon ageing and infection.** (**a**,**b**) Representative images of the adult posterior midgut of *ATF3::GFP* female flies. ATF3::GFP is stained with an anti-GFP antibody (green), DNA is stained with DAPI (blue), Delta-positive intestinal stem cells are red membrane signal (ISC) (white arrow) and red nuclei signal indicates prospero-positive Enteroendocrine cells (EE) (white arrowhead). ATF3::GFP is mainly expressed in large nuclei enterocyte-like cells and Delta-positive ISCs, however not in EE cells. (**c**,**d**) *Act-Gal4; tub-Gal80ts* (*Act*[ts])-driven *ATF3* RNAi (*Act*[ts] > *ATF3-RNAi*) reduces ATF3::GFP expression in the midgut as compared to control *Act*[ts] > *WT* (2 days at 29 °C). (**e**) The relative expression of *ATF3* mRNA in the intestine of *w1118* female flies (wild-type, *WT*) after infection with *P.e* was measured by RT-qPCR. Shown are normalized values. (**f**) The relative expression of *ATF3* mRNA in the intestine of WT female flies upon feeding with 5 mM paraquat was measured by RT-qPCR. Intestines were collected at indicated time points. Mean fold change ± s.e. based on three replicated experiments. The significant differences in gene expression between each challenged group (*P. e* and paraquat treatment) and the unchallenged group are indicated with asterisks (*$P < 0.05$; **$P < 0.01$; ***$P < 0.001$, Student's t-test). (**g**) Representative image of control adult female midgut-*MyoIA-Gal4,UAS-GFP; tub-Gal80ts* (*MyoIA*[ts]) > *WT* and (**h**) ATF3 RNAi midgut-*MyoIA*[ts] > *ATF3-RNAi #1* after 7 days at 29 °C. Intestines were stained for pH3 (red) and DNA (blue), *MyoIA-Gal4* drove *UAS-GFP* expressed in enterocytes and shown in green. (**i**) Quantification of pH3-positive cells per adult midgut of the indicated genotypes (*MyoIA*[ts] > *WT*, *MyoIA*[ts] > *ATF3-RNAi#1*, *MyoIA*[ts] > *ATF3-RNAi#2*, respectively) after 7 days at 29 °C. (**j**) Control WT male midgut and (**k**) *atf3*[76] hemizygous mutant midgut was stained for pH3 (red) and prospero (green) antibodies. (**l**) Quantification of pH3-positive cells per adult midgut of the indicated genotypes (*w1118*-wild-type male, *atf3*[76]-ATF3 mutant male, respectively) after 3 days at 25 °C. *atf3*[76] mutant male midguts contain significantly more mitotic cells than controls. *P* values from Student's t-test are shown in **i** and **l**. Mean ± s.e. Numbers of guts scored for each genotype are indicated from three replicated experiments. Scale bars: 30 μm (**a**,**b**,**e**,**f**,**j**,**k**).

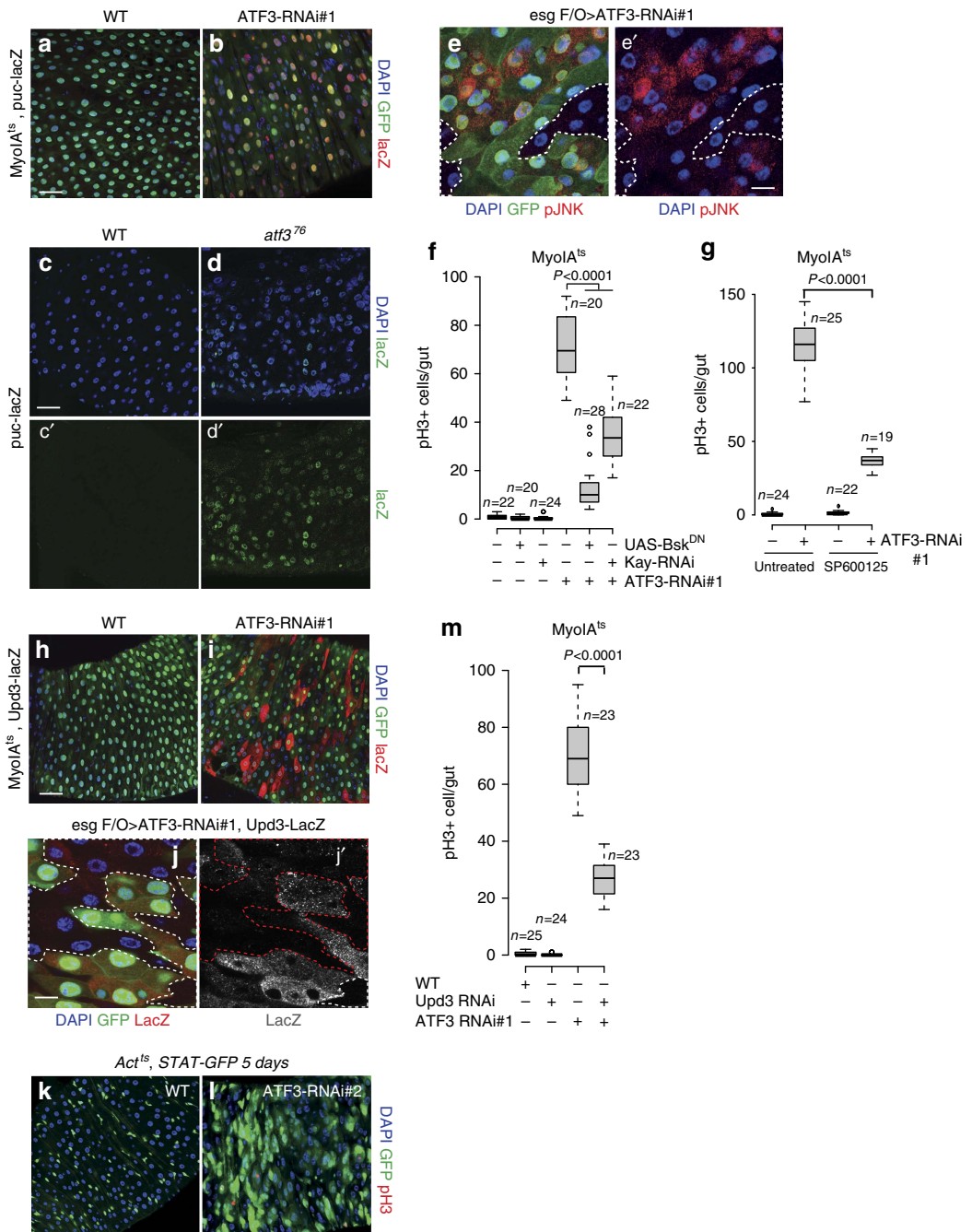

**Figure 2 | ATF3 restricts ISC division by regulation of JNK and STAT signalling.** (**a–d**) Induction of *puckered-lacZ* (*puc-lacZ,* red) in the female midgut epithelium after RNAi ATF3 in enterocytes (*MyoIA*$^{ts}$*>ATF3-RNAi#1*) for 3 days at 29 °C. DNA are stained with DAPI in blue and *MyoIA-GFP* are shown in green. (**c,d**) WT adult male midgut (**c-c'**) and *atf3*$^{76}$ mutant male (**d-d'**) expressing *puckered-lacZ* (green) and DNA are stained with DAPI (blue). (**e**) The activation of JNK (red) in *esg*$^{ts}$ F/O (*esg-Gal4,UAS-GFP; Act>STOP>Gal4, tub-Gal80ts,UAS-flp*) induced ATF3 RNAi cell clones after 2 days at 29 °C, monitored by phosphor-JNK (pJNK) staining, *esg*$^{ts}$ F/O-GFP are in green and DNA are stained with DAPI in blue. (**f**) Quantification of pH3-positive cells per adult female midgut of the indicated genotypes (*MyoIA*$^{ts}$*>WT, MyoIA*$^{ts}$*>UAS-Bsk*$^{DN}$*, MyoIA*$^{ts}$*>Kay-RNAi, MyoIA*$^{ts}$*>ATF3-RNAi#1, MyoIA*$^{ts}$*>ATF3-RNAi#1&UAS-Bsk*$^{DN}$*, MyoIA*$^{ts}$*>ATF3-RNAi#1 and Kay-RNAi,* respectively) after shift to 29 °C for 7 days. (**g**) Quantification of pH3-positive cells per adult midgut of 7 days old WT and ATF3 RNAi female flies cultured on 5% sucrose filter paper discs and treated with or without the JNK inhibitor SP600125 for additional 3 days at 29 °C. (**h,i**) Induction of the *Upd3-lacZ* (red) in the midgut epithelium after ATF3 RNAi in enterocytes (*MyoIA*$^{ts}$*>ATF3-RNAi#1*) for 3 days at 29 °C, DNA are stained with DAPI in blue and *MyoIA-GFP* are shown in green. (**j**) *Upd3-lacZ* expression (red in **j** and grey in **j'**) in *esg*$^{ts}$ F/O induced ATF3 RNAi cell clones after 2 days at 29 °C. (**k,l**) Representative images of *Act*$^{ts}$*>WT* (**k**) and *Act*$^{ts}$*>ATF3-RNAi#2* (**l**) female midguts carrying a GFP reporter for JAK/STAT activity (*10XSTAT-GFP,* green) after 5 days at 29 °C, intestines were stained for pH3 (red) and DNA (blue). (**m**) pH3 quantification per female midgut of flies with indicated genotypes (*MyoIA*$^{ts}$*>WT, MyoIA*$^{ts}$*>Upd3-RNAi, MyoIA*$^{ts}$*>ATF3-RNAi#1, MyoIA*$^{ts}$*>ATF3-RNAi#1 and Upd3-RNAi,* respectively) shifted to 29 °C for 7 days. Upd3 is required for ATF3 loss induced ISC proliferation. *P* values from Student's *t*-test are shown in **f,g** and **m**. Mean ± s.e. Numbers of guts scored for each genotype are indicated from three replicated experiments. Scale bars: 30 µm (**a-e,h-l**).

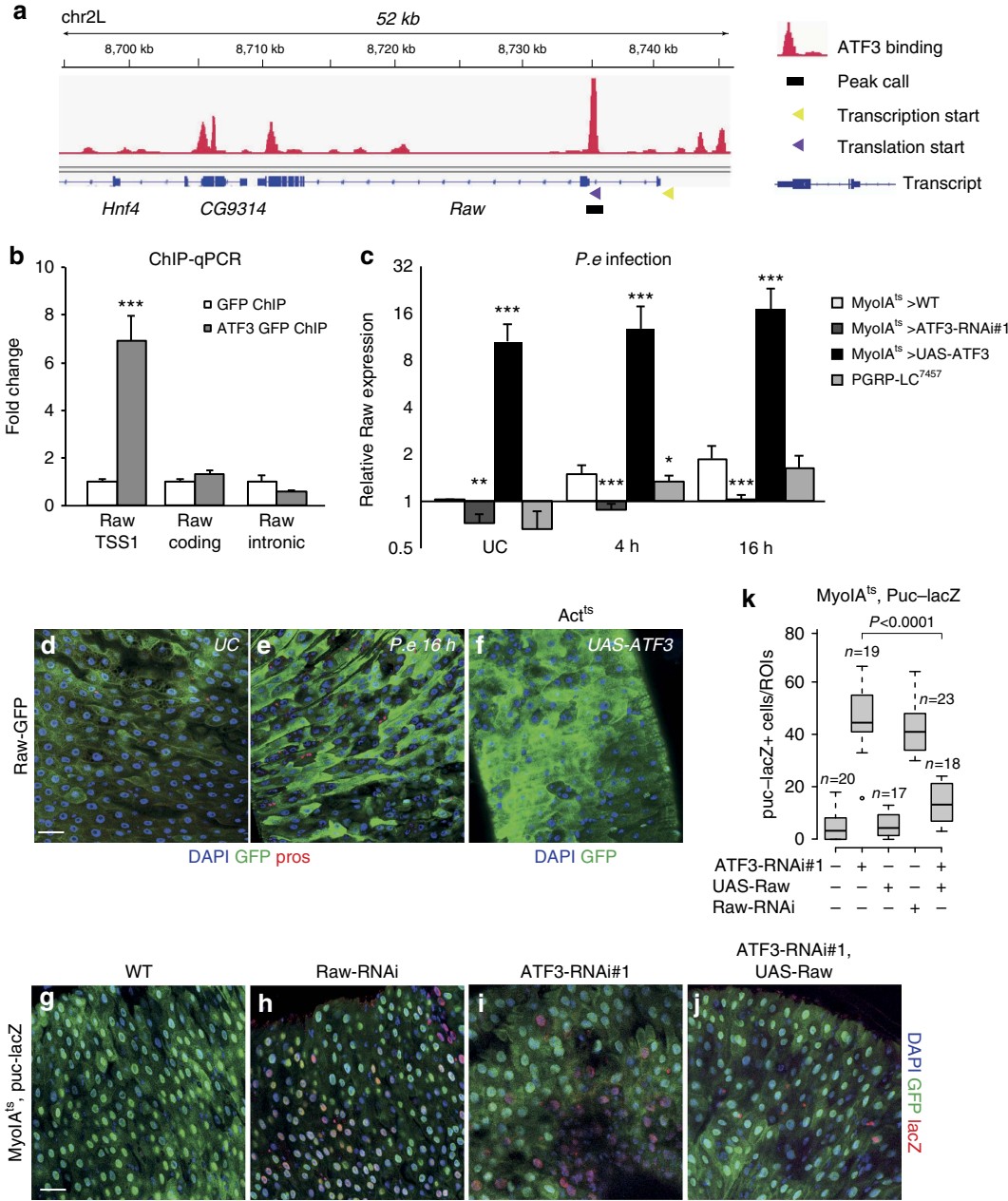

**Figure 3 | ATF3 transcriptionally regulates *Raw* to restricts JNK activity.** (**a**) ChIP-seq track for ATF3-GFP protein at the Raw locus. Black block represents bound region (peak enriched region). ATF3 binds at the first intron of *Raw*. (**b**) ChIP-qPCR analysis at *Raw* locus in midgut of *ATF3::GFP* female flies compared to WT controls. A region in the Raw-coding sequence and non-relevant intron region were selected as negative controls. *P* values (**P* < 0.05; ***P* < 0.01; ****P* < 0.001, Student's *t*-test) are shown in **b** and **c**. Mean fold change ± s.e. based on three replicated experiments. (**c**) The relative expression of *Raw* mRNA in the midgut of control (*MyoA^ts^ > WT*), ATF3 knock down (*MyoA^ts^ > ATF3-RNAi*), ATF3 overexpression (*MyoA^ts^ > UAS-ATF3*) and *PGRP-LC^7457^* female flies in unchallenged, 4 or 16 h *P.e* ingestion conditions, assayed by qRT-PCR. The significant differences in gene expression between each RNAi group (*MyoA^ts^ > ATF3-RNAi#1 or UAS-ATF3, and PGRP-LC^7457^, MyoA^ts^ > Raw-RNAi*) and the control group (*MyoA^ts^ > WT*) are indicated with asterisks (**P* < 0.05; ***P* < 0.01; ****P* < 0.001, Student's *t*-test). (**d,e**) Induction of Raw-GFP reporter gene in the midgut epithelium at 0 and 16 h after *P.e* ingestion. Raw-GFP are shown in green and DNA are stained with DAPI (blue). (**f**) Induction of Raw-GFP reporter gene in the midgut epithelium of ATF3 overexpression female flies (*Act^ts^ > UAS-ATF3*), Raw-GFP are shown in green and DNA are stained with DAPI (blue). (**g–j**) The expression of the *puc-lacZ* (red) in the midgut epithelium of *MyoA^ts^ > WT, MyoA^ts^ > Raw-RNAi, MyoA^ts^ > ATF3-RNAi#1* and *MyoA^ts^ > ATF3-RNAi#1,UAS-Raw* female flies for 1 days at 29 °C, MyoA-GFP are shown in green and DNA are stained with DAPI (blue). (**k**) Quantification of *puc-lacZ* positive cells per midgut of female flies with indicated genotypes (*MyoA^ts^ > WT, MyoA^ts^ > ATF3-RNAi#1, MyoA^ts^ > UAS-Raw, MyoA^ts^ > Raw-RNAi, MyoA^ts^ > ATF3-RNAi#1 and UAS-Raw*, respectively) shifted to 29 °C for 1 days. *P* values from Student's *t*-test are shown in **k**. Mean ± s.e. Numbers of guts scored for each genotype are indicated from three replicated experiments. Scale bars: 30 μm (**d–f,g–j**).

infection (Fig. 3c). However, *Raw* mRNA expression increased to a level similar to WT at 16 h post infection, suggesting that the ability of infection to induced *Raw* expression is independent of immune response pathway. In summary, these results indicate that ATF3 transcriptionally upregulates *Raw* expression in the *Drosophila* intestine.

To further explore the function of *Raw* we examined the expression a *Raw-GFP* enhancer trap reporter line (see Methods). Weak expression of *Raw-GFP* was observed exclusively in enterocytes in the midgut (Fig. 3d and Supplementary Fig. 3d), and no expression was detected in Prospero-positive EEs or Delta-positive ISCs (cells with small nuclei) (Fig. 3d and Supplementary Fig. 3e). Moreover, we found *Raw-GFP* expression to be highly induced in enterocytes when the gut was infected with *P.e.* (Fig. 3e). *Raw-GFP* expression was also induced in ATF3 over-expressing intestines (Fig. 3f). Importantly, we observed strong induction of the JNK pathway components, Puc and Kay, in *Raw*-depleted midguts by qPCR (Supplementary Fig. 3f–g). As with *ATF3 RNAi*, we detected enterocyte-specific induction of *Puc-lacZ* after depleting *Raw* using RNAi (Fig. 3g–i). Conversely, *Raw* overexpression reduced the induction of *Puc-lacZ* caused by *ATF3-RNAi* (Fig. 3j,k and Supplementary Fig. 3h-i). These data indicate that *Raw* is a transcriptional target of ATF3 in enterocytes, and that ATF3 and Raw function together as a module to negatively regulate intestinal JNK activity.

**Raw is required to maintain intestinal homeostasis**. Next, we tested whether Raw depletion caused similar defects in intestinal homeostasis as ATF3 depletion. We depleted *Raw* in enterocytes by RNAi and examined effects on ISC mitosis (Fig. 4a,b). Like ATF3 depletion, *Raw-RNAi* in enterocytes increased ISC division (Fig. 4b). Similarly, inactivation of JNK activity using *UAS-Bsk*[DN] suppressed *Raw-RNAi* induced ISC division (Fig. 4b–e).

Previous studies showed that infection leads to intestinal dysplasia with increased numbers of *esg*-positive cells (ISCs and EBs) and a concurrent increase in Armadillo (Arm) staining (a marker of ISC:EB adhesion junctions)[11]. To find out whether loss of ATF3 or Raw causes a similar dysplastic phenotype, we stained depleted midguts for Arm. ATF3- or Raw-depleted midguts showed dramatically increased number of small cells with high Arm signals (Supplementary Fig. 4a–f). This phenotype is similar to infection-induced intestinal dysplasia[11], indicating that depletion of ATF3 or Raw perturbs intestinal homeostasis and leads to intestinal dysplasia.

**Raw acts downstream of ATF3 to control JNK activity**. To further assess the epistatic relationship between ATF3 and Raw, we analysed the effects of either Raw gain-of-function under ATF3 depletion conditions or Raw knockdown in ATF3 over-expression conditions. We found that overexpression of Raw suppressed ATF3 depletion-induced hyper-proliferation (Fig. 4f). Conversely, depletion of Raw caused strong increased in mitoses in the intestine, even in the presence of over-expressed ATF3 (Fig. 4g). In addition, we observed that heterozygosity for a mutation in *puckered* (*puc*), a known JNK antagonist, enhanced *ATF3* or *Raw* RNAi induced ISC mitosis (Supplementary Fig. 4g). In contrast, overexpression of puc showed no reduction in ATF3 RNAi induced ISC proliferation (Supplementary Fig. 4h). These results suggest that Raw is epistatic to ATF3, which is independent of puc, to control JNK activity in the intestine.

To further explore whether ATF3 and Raw act in a feedback loop to control JNK activity in enterocytes, we altered JNK signalling by expressing either an active form of JNKK (*Hep*[Act]) or a dominant negative form of JNK (*Bsk*[DN]). We then examined *ATF3* and *Raw* expression in the presence or absence of *P.e* infection. As expected, JNK activation induced *puc* expression, whereas *puc* expression was reduced upon JNK suppression (Supplementary Fig. 4i). We observed a strong induction of *ATF3* and *Raw* expression in JNK hyperactivated midguts (Supplementary Fig. 4j–k). However, blocking JNK activity did

not prevent the induction of *ATF3* and *Raw* upon *P. e* infection (Supplementary Fig. 4j–k), suggesting that JNK signalling is not required for ATF3 and Raw induction by stress.

Consistent with the results we had obtained using *ATF3-RNAi*, we found a significant upregulation of *Upd3* and *Socs36E* mRNA levels in response to *Raw-RNAi* expression (Supplementary Fig. 4l–m). Depletion of *Raw* also resulted in an activation of STAT signalling as assayed using the *Upd3-lacZ* and *STAT-GFP* reporters (Fig. 4h–k and Supplementary Fig. 4n–p). In addition, *Upd3 RNAi* strongly suppressed the induction of ISC proliferation caused by *Raw* depletion (Fig. 4l). These results all suggest that loss of *Raw* induces intestinal dysplasia by de-repressing intrinsic JNK activity, and thereby leading to activation of JAK/STAT signalling to promote stem cell proliferation.

**ATF3-depleted flies are susceptible to infection**. ATF3 is known to be involved in inflammatory and stress signalling in various organisms[32–34]. We hypothesized that ATF3 has a protective role in the *Drosophila* intestine in response to bacterial infection. We therefore analysed the role of ATF3 in the resistance to oral infection with *P.e*. ATF3-deficient flies were more susceptible to *P.e.* infection as compared to wild-type controls (Fig. 5a). The level of susceptibility of ATF3 depleted flies was similar to that observed for Imd pathway deficient flies (*PGRP-LC*[7457], see Methods) (Fig. 5a). Conversely, flies in which ATF3 was overexpressed in gut enterocytes were more resistant to *P.e.* infection than wild-type controls (Fig. 5a). A similar enhancement of survival was observed in Raw-overexpressing flies (Fig. 5b). Moreover, overexpression of Raw rescued the susceptibility to *P.e.* infection caused by ATF3 depletion (Fig. 5b).

Interestingly, we noticed a reduction in the length of ATF3 depleted midguts following *P.e.* infection (Fig. 5e and Supplementary Fig. 5a), while no significant change was observed on the morphology of ATF3 overexpressing midguts (Fig. 5c,d and Supplementary Fig. 5a). However, a significant increase in numbers of mitotic cells was observed in both ATF3- and Raw-deficient midguts after infection, as compared to controls (Fig. 5f). Conversely, overexpression of either ATF3 or Raw significantly reduced the ISC proliferation induced by bacterial infection (Fig. 5g–m). Consistent with this, we also observed that the expression of signalling components required for intestinal regeneration (*Upd2*, *Upd3*, *Socs36E*, *Spi* and *puc*) was down-regulated in ATF3 over-expressing midguts (Supplementary Fig. 5b). These results suggest that ATF3 and Raw have a protective role in enterocytes in infection conditions. In addition, they raise the interesting question of why more stem cell proliferation would be associated with reduced survival upon infection. Previous reports have nearly all concluded that stem cell proliferation is an essential aspect of epithelial damage repair following infection[7,10,23,44].

**ATF3 and Raw modulate apoptosis via the JNK activity**. The hyperplasia and homeostasis defects observed in ATF3-deficient midguts are similar to those resulting from tissue damage, for instance from chemical toxins or bacterial infection[9,10]. Hence we hypothesized that the physiological role of ATF3 in enterocytes may be to protect cells from apoptosis or environmentally induced stress. JNK signalling can induce apoptosis via upregulating pro-apoptotic genes, or by affecting the activity of pro- and anti-apoptotic proteins through phosphorylation[45]. In the fly midgut, activation of JNK in enterocytes can cause cell death and stem cell proliferation[10]. Therefore, we assayed dying cells in *Raw-RNAi* or *ATF3-RNAi* expressing midguts using either anti-cleaved Death Caspase 1 (DCP1) or TUNEL staining. Indeed, after depleting Raw or ATF3 we observed high levels of

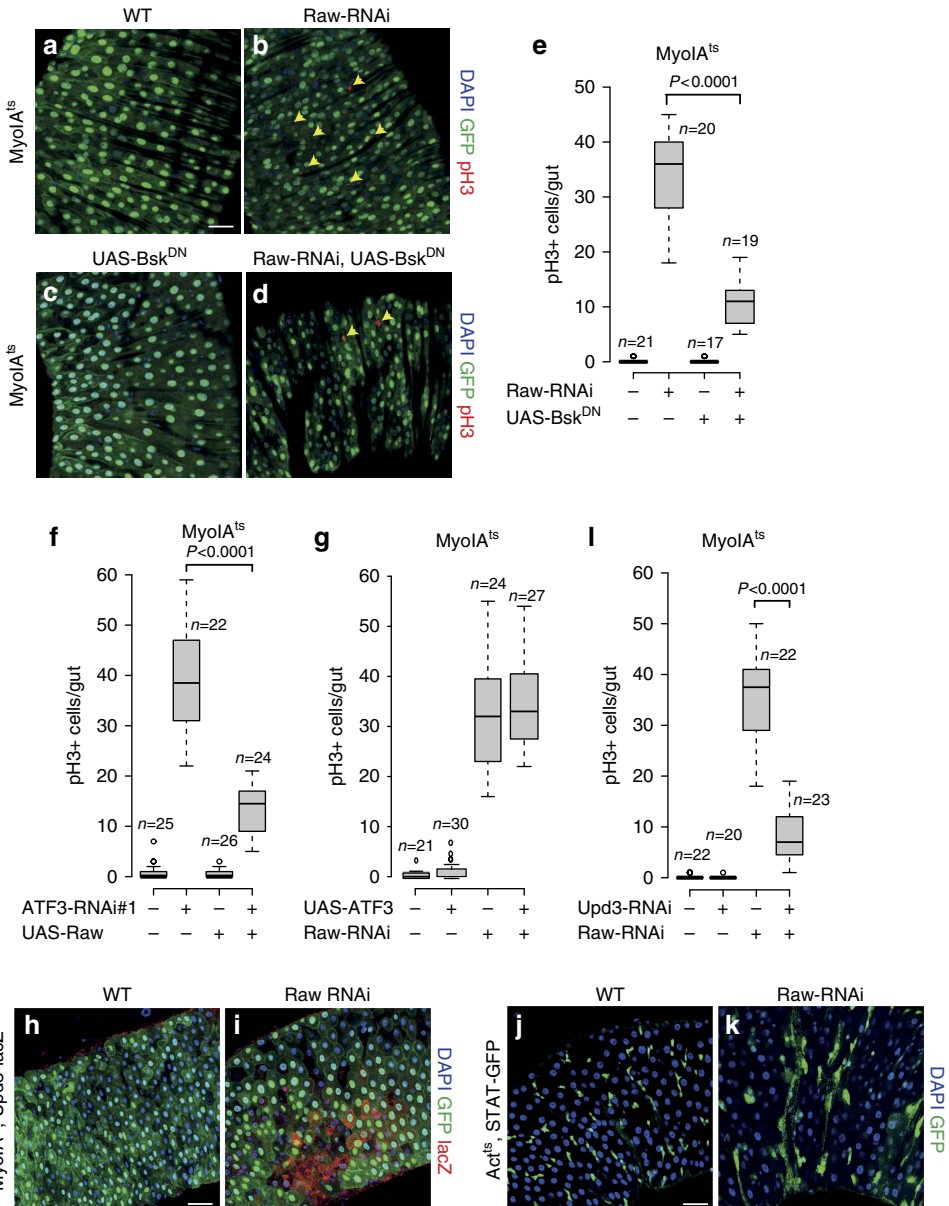

**Figure 4 | Raw is epistatic to ATF3 to restrict ISC proliferation.** (**a–d**) Representative posterior midguts image of phosphor-Histone 3 (pH3) staining of *MyoIA^ts > WT*, *MyoIA^ts > UAS-Bsk^DN*, *MyoIA^ts > Raw-RNAi*, *MyoIA^ts > Raw-RNAi,UAS-Bsk^DN* female flies for 5 days at 29 °C. (**e**) pH3 Quantification per midgut of female flies with indicated genotypes (*MyoIA^ts > WT*, *MyoIA^ts > Raw-RNAi*, *MyoIA^ts > UAS-Bsk^DN*, *MyoIA^ts > Raw-RNAi* and *UAS-Bsk^DN*, respectively) shifted to 29 °C for 5 days. (**f**) pH3 Quantification per midgut of female flies with indicated genotypes (*MyoIA^ts > WT*, *MyoIA^ts > ATF3-RNAi#1*, *MyoIA^ts > UAS-Raw*, *MyoIA^ts > ATF3-RNAi#1 and UAS-Raw*, respectively) shifted to 29 °C for 5 days. (**g**) pH3 Quantification per midgut of female flies with indicated genotypes (*MyoIA^ts > WT*, *MyoIA^ts > UAS-ATF3*, *MyoIA^ts > Raw-RNAi*, *MyoIA^ts > Raw-RNAi* and *UAS-ATF3*, respectively) shifted to 29 °C for 5 days. (**h,i**) Induction of the *Upd3-lacZ* (red) in the midgut epithelium after Raw RNAi in enterocytes (*MyoIA^ts > Raw-RNAi*) for 1 day at 29 °C, DNA are stained with DAPI in blue and *MyoIA-GFP* are shown in green. (**j,k**) Representative images of *Act^ts > WT* (**j**) and *Act^ts > Raw-RNAi* (**k**) female midguts carrying a GFP reporter for JAK/STAT activity (10X*STAT-GFP*, green) after 1 day at 29 °C, DNA are stained with DAPI (blue). (**l**) pH3 Quantification per midgut of female flies with indicated genotypes (*MyoIA^ts > WT*, *MyoIA^ts > Upd3-RNAi*, *MyoIA^ts > Raw-RNAi*, *MyoIA^ts > Raw-RNAi* and *Upd3-RNAi*, respectively) shifted to 29 °C for 5 days. *P* values from Student's *t*-test are shown in **e,f,g** and **l**. Mean ± s.e. Numbers of guts scored for each genotype are indicated from three replicated experiments. Scale bars: 30 µm (**a–d,h–k**).

cell death in most enterocytes (Fig. 6a,c,d and Supplementary Fig. 6a–d). High enterocytes apoptosis was also observed in *P.e.* infected midguts (Fig. 6b), which is consistent with previous findings[11]. Conversely, overexpression of Raw suppressed ATF3 depletion-induced enterocyte apoptosis (Fig. 6d–f). To further explore the role of ATF3 and JNK in the regulation of enterocyte apoptosis, we overexpressed *Hep^Act* in enterocytes in combination with an additional copy of ATF3 (*ATF3::GFP*) or overexpression

of ATF3 (*UAS-ATF3*) (Fig. 6g–k and Supplementary Fig. 6e–i). Inducing *Hep^Act* expression in enterocytes for 4 days caused massive cell death in midguts, and resulted in a shorter (shrunken) midgut (Fig. 6g,h and Supplementary Fig. 6e). Introduction of an additional copy of *ATF3* (*ATF3::GFP*) partially rescued the 'shrunken gut' phenotype (Supplementary Fig. 6f), and co-expression of *ATF3* with *Hep^Act* strongly suppressed JNK-mediated mitosis and apoptosis in the midguts

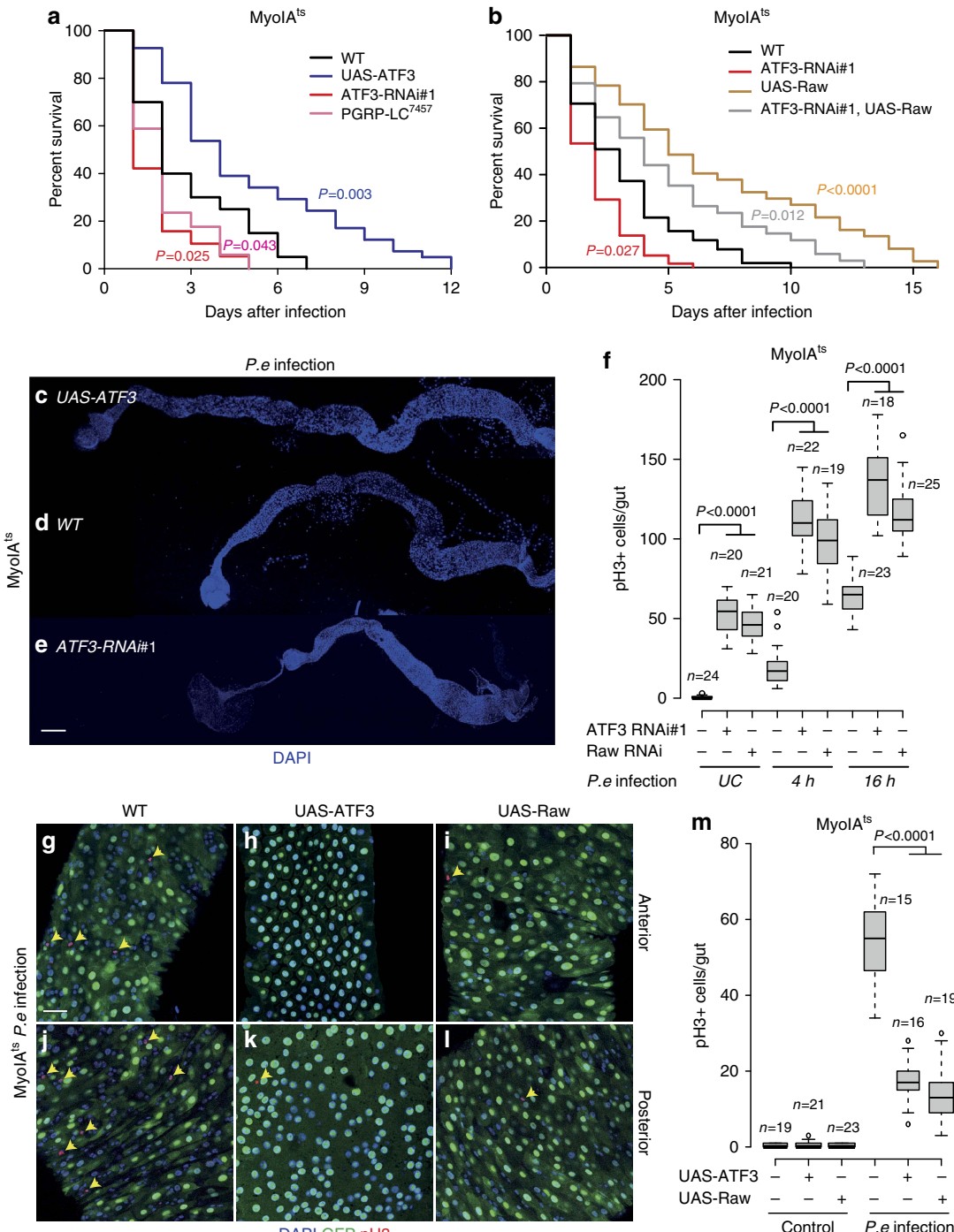

**Figure 5 | A protective role of ATF3-Raw for survival against *P. e* infection.** (**a**) A survival analysis of flies orally infected with the pathogenic bacteria *P. e* revealed an increased susceptibility of female ATF3 RNAi (*MyoA^ts^>ATF3-RNAi#1*) flies, whereas ATF3 overexpression (*MyoA^ts^>UAS-ATF3*) leads to increased survival upon infection. (**b**) A survival analysis of flies orally infected with the pathogenic bacteria *P.e* revealed an increased survival of Raw overexpression female flies (*MyoA^ts^>UAS-Raw*); likewise, Raw overexpression rescued ATF3 depletion (*MyoA^ts^>ATF3-RNAi#1, UAS-Raw*) induced susceptibility to *P.e* infection. *P* values from LogRank test are shown in **a** and **b**. (**c–e**) The representative image of DAPI staining of whole midgut of adult female *Drosophila* (*MyoA^ts^>UAS-ATF3; MyoA^ts^>ATF3-RNAi#1, MyoA^ts^>WT*) after *P. e* ingestion for 20 h. The gut length of *MyoA^ts^>ATF3-RNAi* flies is significantly shorter than *MyoA^ts^>UAS-ATF3* and *MyoA^ts^>WT* flies. (**f**) pH3 quantification per midgut of flies with indicated genotypes (*MyoA^ts^>WT, MyoA^ts^>ATF3-RNAi#1, MyoA^ts^>Raw-RNAi*, flies are either in unchallenged-UC, 4 h or 16 h *P.e* ingestion conditions respectively) shifted to 29 °C for 5 days and then feed with *P.e*. (**g–l**) Representative image of pH3-positive mitotic cells in 2 days old of control (*MyoA^ts^>WT*) and *MyoA^ts^>UAS-ATF3* and *MyoA^ts^>UAS-Raw* female midguts in response to *P.e* infection. (**m**) Quantification of pH3-positive cells per adult female midgut of the indicated genotypes after shift to 29 °C for 2 days and then flies were fed with *P.e* for 16 h. *P* values from Student's *t*-test are shown in **f** and **m**. Mean ± s.e. Numbers of guts scored for each genotype are indicated from three replicated experiments. Scale bars: 250 μm (**c–e**), 30 μm (**g–l**).

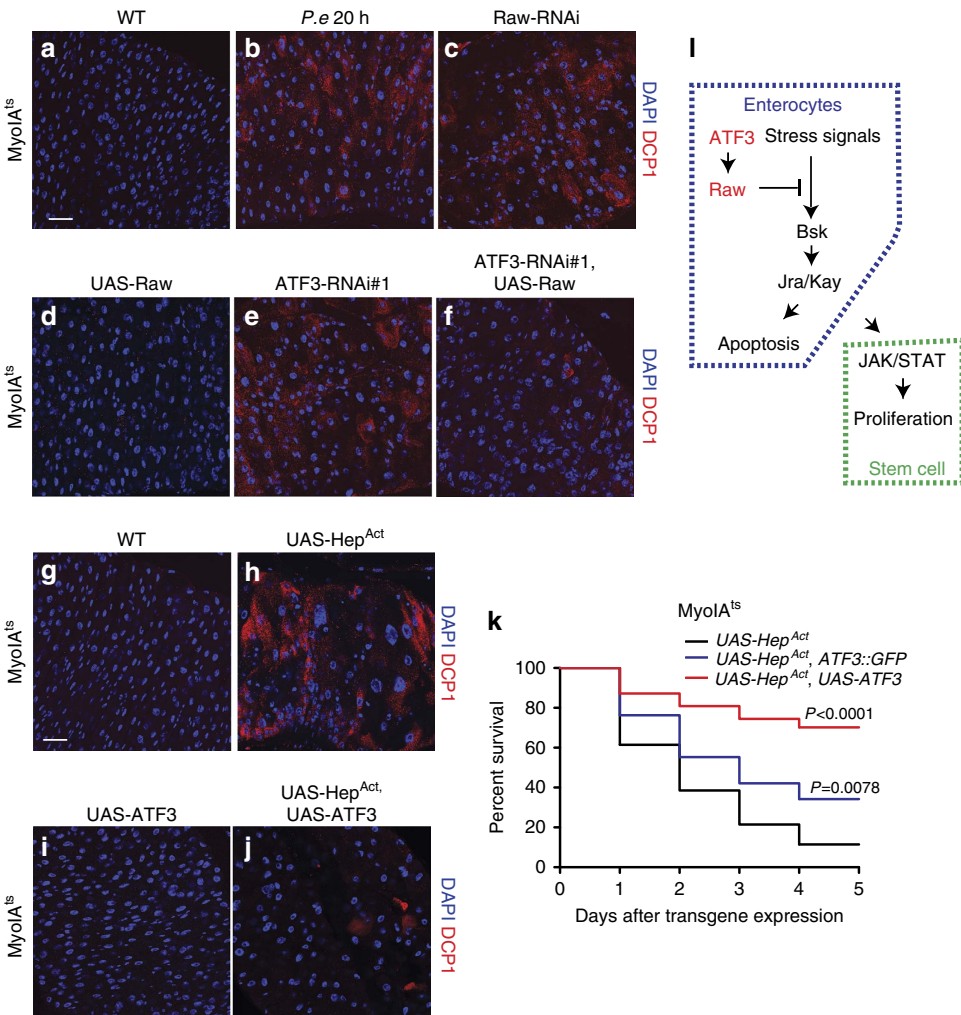

**Figure 6 | ATF3-Raw controls JNK mediated apoptosis.** (**a–f**) Representative image of anti-cleaved Death Caspase1 (DCP1) staining in 5 days old female of control (*MyoA^ts^ > WT*), *P.e* infected control (*MyoA^ts^ > WT*), *MyoA^ts^ > Raw-RNAi*, *MyoA^ts^ > UAS-Raw*, *MyoA^ts^ > ATF3-RNAi#1* and *MyoA^ts^ > ATF3-RNAi#1 and UAS-Raw* posterior midguts. DCP1-positive cells are in red and DNA are stained with DAPI in blue. (**g–j**) Representative image of anti-cleaved Death Caspase1 (DCP1) staining in 2 days old female of control (*MyoA^ts^ > WT*), *MyoA^ts^ > UAS-Hep^Act^*, *MyoA^ts^ > UAS-ATF3*, and *MyoA^ts^ > UAS-Hep^Act^ and UAS-ATF3* posterior midguts. DCP1-positive cells are in red and DNA are stained with DAPI in blue. Expression of ATF3 suppressed JNK hyper-activation induced apoptosis. (**k**) Introducing additional copy of ATF3 or overexpression of ATF3 rescue JNKK (*Hep^Act^*) hyperactivation induced lethality. *P* values from LogRank test are shown in **k**. (**l**) An intermediate model of ATF3-Raw regulates JNK mediated enterocytes apoptosis and ISC proliferation. Scale bars: 30 μm (**a-f**, **g-j**).

(Fig. 6i,j and Supplementary Fig. 6g–i). Importantly, overexpression of ATF3 rescued the lethality caused by hyperactivation of JNK activity (Fig. 6k). This suggests that ATF3 is required in the control of JNK-mediated enterocytes apoptosis. Hence, our data indicate that ATF3-Raw control JNK mediated regeneration and cell death (Fig. 6l).

To further support these findings, the caspase inhibitor *p35* was co-expressed along with *ATF3-RNAi* in enterocytes (Supplementary Fig. 6j–n). Expression of *p35* was able to greatly reduce the increase in mitotic cells caused by depletion of ATF3 (Supplementary Fig. 6m-n). Consistent with our findings, overexpression of *p35* in intestinal epithelial cell has been shown to suppress bleomycin-induced ISC proliferation[19]. A tight control of cell proliferation, differentiation and death is required to maintain tissue homeostasis, whereas dysregulation of apoptosis causes imbalance in homeostasis and harmful effects on organismal health[46]. Hence, we speculate that JNK activity is enhanced in the absence of ATF3, and that this induces high levels of enterocytes cell death that cannot be compensated by the extra stem cell proliferation. This imbalance could lead to morphological changes and a 'shrunken' gut phenotype.

**Susceptibility to infection is due to intestinal barrier dysfunction.** To further explore how ATF3 depletion led to susceptibility to infection, we assessed intestinal barrier dysfunction using the 'Smurf' assay[47]. Upon feeding with a blue dye mixed with concentrated *P.e*, flies expressing *ATF3-RNAi* showed a high incidence of intestinal barrier dysfunction compared to wild-type controls (Fig. 7a–c). Conversely, overexpression of *ATF3* in enterocytes resulted in reduced Smurf positivity and better survival following *P.e.* infection (Fig. 7a–c).

To further test our hypothesis that ATF3 buffers JNK activity to control enterocytes apoptosis and maintain tissue integrity during infections, we suppressed JNK activity (Bsk^DN^) in *ATF3-RNAi* expressing flies and then assessed the incidence of smurf-positive flies upon infection. Indeed, we observed better survival following infection and a lower incidence of intestinal

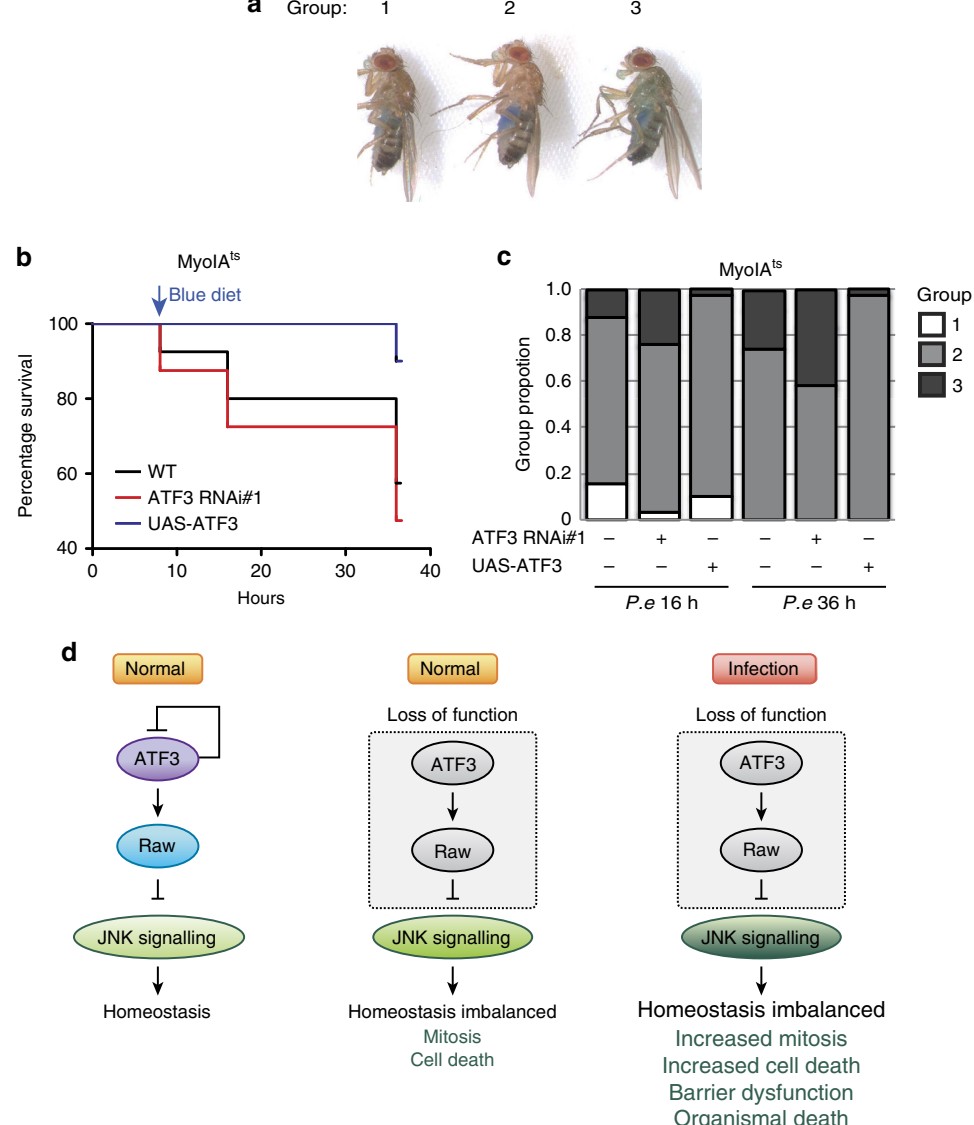

**Figure 7 | ATF3 loss caused susceptibility is associated with barrier dysfunction.** (**a**) Images of Smurf flies, representing different amount of blue dye in the haemolymph and divided into 1, 2 and 3 groups. (**b**) A parallel survival analysis of orally *P.e* infected female flies. Flies were fed with *P.e* diet (5% sucrose) for 8 h and transferred to Acid blue *P.e* diet and survival rate is monitored at 8, 16 and 36 h respectively. (**c**) The proportion of Smurf flies in each group following 8 h or 24 h transfer to blue food. *n* = 40 flies per genotype per experiment. Three independent experiments have been performed. (**d**) Predicted model for ATF3-Raw regulates JNK activity in homeostatic and stress conditions.

barrier dysfunction in *Myots > UAS-Bsk^DN, ATF3-RNAi* flies, than in controls expressing *ATF3-RNAi* alone (Supplementary Fig. 7a–c). These results suggest that ATF3-depletion after infection induces excessive JNK activity, which results in hyperproliferation of ISCs but simultaneously leads to massive enterocytes cell death and intestinal barrier dysfunction, which eventually disrupts homeostasis and results in organismal death (Fig. 7d and Supplementary Fig. 8).

## Discussion

Breakdown of intestinal homeostasis has been linked to many diseases as well as premature ageing[48,49]. After enteric infection, mechanisms have to be in place to restore gut epithelial homeostasis by mounting an immune response, clearing damaged cells, and regenerating the compromised epithelium. Stress signalling and immune response pathways have to be coordinated with regenerative responses, and over-activation of pro-apoptotic signals must be avoided.

ATF3 is a stress-inducible mediator of cellular stress response signalling[32]. In this study, we show that ATF3 plays a central role in the regulation of apoptotic responses and tissue regeneration following bacterial infection. ATF3 depletion in the *Drosophila* intestine resulted in inappropriate JNK activation, and thereby increased epithelial cell death and stem cell proliferation. We provide several lines of evidence that illustrate how ATF3 restricts ISC proliferation and balances stress responses with regeneration. First, we found that ATF3 antagonizes JNK activity to restrain gut epithelial cell death and stem cell activity. Second, we discovered that ATF3 modulates JNK activity through the transcriptional control of *Raw*, which negatively regulates JNK signalling[42,43]. Third, we show that this ATF3-Raw module restrains JNK-mediated apoptosis in gut enterocytes. Strikingly, ATF3- or *Raw*-overexpression reduced

the loss of intestinal epithelium integrity following *P.e.* infection, and thereby enhanced resistance to this enteric pathogen.

ATF3 has been shown to regulate genes involved in metabolism, cell cycle, apoptosis, cell adhesion and signalling, including insulin, p53, Wnt and VEGF pathways[50]. Moreover, ATF3 functions in the fly larval intestine to regulate metabolic and immune homeostasis[35]. Chakrabati *et al.* reported that ATF3 plays a role in lipid metabolism and that its expression is regulated by the p38c MAP kinase pathway, a known stress- and immune-response pathway[34,51]. Given the fact that ATF3 is induced by paraquat, this suggests that ATF3 expression may be regulated by p38c signalling upon oxidative stress. In addition, we found ATF3 may auto-regulate itself transcriptionally by binding to its own promoter. A previous study showed that the expression of ATF3 is upregulated in *atf3* mutants[35]. Consistently, ATF3 auto-repression of its own transcription has been documented in both cultured human cells and mouse brain[52,53]. Hence, the regulation of ATF3 expression in the fly's intestine appears to be complex and context-dependent. However, further studies might be necessary to dissect the auto-regulatory role of ATF3 at a mechanistic level.

ATF3 is induced in most cells after various stress signals, including DNA damage, endoplasmic reticulum stress, oxidative stress, infection and carcinogen exposure[54]. For instance, Reactive Oxygen Species (ROS) strongly induces ATF3 expression and protect mice against endotoxic shock[55]. Several studies in mice showed ATF3 regulates TLR-associated inflammatory responses and bacterial infection induces ATF3 expression[56–59]. Together with our study, these observations indicate a conserved role of ATF3 as a stress response gene.

We demonstrate here that ATF3 controls JNK activity through transcriptional regulation of *Raw* expression. The *Drosophila* dorsal-open group gene *Raw* is known to be required for restricting JNK signalling during embryogenesis[42,60]. Byars *et al.* observed elevated JNK activity in *Raw* mutant embryos[60]. Further genetic evidence showed that *Raw* acts upstream to Bsk (JNK) and functions as an antagonist of the JNK signalling response to oxidative stress[41]. A recent study showed that a non-coding RNA, *acal*, functions downstream of *Raw* to control the expression of Cka, a conserved protein that forms a complex with Bsk to phosphorylate and activate AP1 (Jra/Kay)[61]. Based on these observations, we suggest that ATF3 regulates *Raw* expression to control *acal* mediated Cka expression, which is required for Jra/Kay activity. Given that JNK signalling is a major effector of gut epithelial cell death and regeneration, it is possible that ATF3, Raw and Puckered function in parallel to control JNK activity. However, other recent studies show that Raw is a transmembrane protein that regulates cell adhesion via other junctional proteins such as Tricornered and Integrins[43,62]. Since adhesion is also critical for enterocytes survival and JNK suppression in the gut epithelium, an adhesion function for Raw might also explain some of the effects we report.

Enteric infection with pathogenic bacteria has profound effects on *Drosophila* survival. Ingestion of *P.e.*, for instance, often results in irreversible damage to the *Drosophila* intestine, killing the host because it prevents tissue repair and renewal[63]. JNK activity can be induced by bacterial and host-produced ROS, and can protect intestinal epithelial cells from oxidative stress by controlling expression of the antioxidant enzyme Peroxiredoxin V (ref. 64). Strong activation of JNK can also trigger apoptosis of enterocytes, and promote ISC proliferation to replenish the damaged apoptotic enterocytes[65]. However, aberrantly high or prolonged JNK signalling can lead to dysplasia involving abnormal excess ISC proliferation and mis-differentiation in aged tissues[30] and therefore JNK signalling has to be tightly regulated since it can influence many biological processes (for example, immune responses, ROS, ISC proliferation and cell death). In this study, we demonstrate that the ATF3-Raw module buffers excessive activation of JNK signalling and thereby restrains epithelial cell death, a function that is essential for normal gut homeostasis and optimal survival following infection. In line with this, JNK hyper-activation leads to massive enterocytes apoptosis, dramatically changes gut morphology, and often results in the organismal death in *Drosophila*. Interestingly, ATF3 depletion in enterocytes enhanced ISC mitosis. However, this increased ISC proliferation failed to compensate massive infection-induced epithelial cell loss, which altered gut morphology and disrupted barrier function. Eventually, ATF3 deficient flies could not maintain proper tissue function or homeostasis and perished. In addition, we observed that the expression of many antimicrobial peptides is dependent on ATF3. ATF3 deficiency can also lead to a decrease of the production and secretion of antimicrobial peptides, which contribute to intestinal epithelium defence mechanism in response to infection (data not shown). However, *Raw* expression rescued the susceptibility to infection caused by ATF3 depletion, suggesting that buffering intestinal JNK activity is critical for tissue regeneration and preventing organismal death. Hence, we conclude that depletion of ATF3 induced massive epithelial cell loss followed by intestinal barrier breakdown, and that this is the primary cause of fly susceptible to infection. It should prove interesting to investigate similar roles for ATF3 in the human and mouse intestine. Although *Raw* has no clear human orthologue, its extracellular domain does show similarities to conserved domains in mucins and leucine rich repeat proteins[62], which could be similarly important in maintaining barrier function in vertebrate endothelia. Cytoprotective JNK signalling influences ISC proliferation and causes accumulation of misdifferentiated ISC progenitors in aged intestines[66]. In addition, JNK signalling has deleterious effects on longevity and is activated during ageing, whereas limiting JNK activation extends lifespan[67]. As ATF3 regulates *Raw* expression during ageing, it would be also interesting to explore the role of ATF3 and Raw on regulating JNK signalling in the ageing processes.

## Methods

**Fly stocks.** The following *Drosophila melanogaster* stocks were used in this study: UAS-Atf3, Atf3::GFP and *atf3*[76] (Mirka Uhlirova, Cologne University); *PGRP-LC*[7457] (PGRP-LC mutant, gift from Akira Goto, INSERM, CNRS); 10X-STAT-GFP (STAT reporter, a gift from Norbert Perrimon); Delta-lacZ, MyoIA-lacZ (Bruce Edgar), Raw-GFP as an enhancer trap reporter (BLN44702), ATF3-GFP (BLN42263) (Bloomington Stock Center). The following RNAi lines were obtained from the Bloomington Stock Center: UAS-ATF3 RNAi#2 (BLN26741), UAS-Raw RNAi (BLN31393), UAS-Upd3 RNAi (BLN32859), UAS-Kay RNAi (BLN31322) and UAS-Yki RNAi (BLN34067). The following RNAi lines were obtained from the Vienna *Drosophila* RNAi Center: UAS-ATF3 RNAi#1 (KK113077), UAS-Upd3 RNAi (KK110348). UAS-ATF3-HA (F000660), UAS-Raw (F001421), UAS-Raw-HA (F001105) were obtained from FLYORF Zurich. When performing *in vivo* experiment, female Gal4 drivers were used to cross with male RNAi or overexpression line and their progenies were selected for genotype and randomly chosen to be used in the experiment. Female adult *Drosophila* were used in all analyses (image, quantification and survival test) throughout the study except *atf3*[76] mutant-related experiments. See Supplementary Table 1 for details of mutant used in this study.

**Infection experiments.** For bacterial feeding and survival experiments, flies were maintained at 18 °C. Adults were collected 3-5 days after eclosion and shifted to 29 °C to induce the RNAi or transgene expression. In feeding experiments, flies were starved at 29 °C for 4 h before transfer to cages containing a filter paper with 1 ml high-dose bacteria (OD600 = 200). Infection and bacterial feeding were performed at 29 °C. For viability analysis, we used 40 flies per vial and the experiments were repeated six times. Flies were transferred to new cages with contaminated filter paper discs every 24 h and counted every day, as previously described[9]. The statistical significances (*P* values) between overexpression or RNAi and control group were performed using LogRank tests with GraphPad Prism5.

**Smurf assay.** The intestinal barrier dysfunction assay was conducted using female flies after shifting to 29 °C for 15 days. The infection experiment has been carried out as described above. Briefly, 40 flies was used per vial and kept at 29 °C. Flies were first exposed to contaminated filter paper for 8 h and then transferred to new cages with contaminated filter paper mixed with 1% Acid Blue[47,68]. The filter paper contained blue diet was renewed once at 16 h after infection. The smurf flies were counted at 16 and 36 h after infection and dead flies were removed during the filter paper renewing. Three independent experiments were performed.

**RT-qPCR.** Ten female midguts were used for RNA extraction using an RNeasy kit (Qiagen). cDNA was sythesized as described above using 1 μg of RNA. RT-qPCR was performed in a 384-well format using the Universal Probe Library and a LightCycler 480 (Roche). All experiments were performed in duplicate, each with three independent biological replicates. All results are presented as the means ± s.d. of the biological replicates[69]. Rp49 was used as an internal control. See Supplementary Table 2 for primer sequences and probes.

**Histology.** For immunofluorescence staining, intestines were dissected in Schneider's medium and fixed in 4% paraformaldehyde (PFA) for 45 min. Tissue were washed with 0.1% Triton-X100 in 1× phosphate-buffered saline (PBS) (PBT), and blocked in 3% of goat serum in 1× PBT for 1 h. Subsequently, tissues were stained with mouse monoclonal anti-Delta (1:100) (C594.9B, Developmental Studies Hybridoma Bank, DSHB), anti-Prospero (1:100) (MR1A, DSHB), anti-Armadillo (1:200) (N2-7A1, DSHB), mouse anti-beta galactosidase (1:1,000) (40-1a, DSHB); rabbit polyclonal anti-cleaved Death Caspase1 (1:600) (Cell Signaling, Catalog Nr. 9578), rabbit polyclonal anti-phosphoSer10-Histone 3 (1:600) (Cell Signaling, Catalog Nr. 9701) and rabbit polyclonal anti-GFP (1:500) (Cell Signaling, Catalog Nr. 2555). More detailed information is provided in Supplementary Table 3. For TUNEL assay, the ApopTag RED *In Situ* Detection Kit (Millipore S7165) was used.

**Phospho-Histone H3 (pH3) analysis.** The mitotic indices of all indicated genotypes and conditions were assessed by counting the dividing cells marked by pH3 staining. For each experiment, 8–10 midguts were dissected. Three independent experiments were performed. Only no autofluorescence and intact tissue were counted and showed in the figure. The data are presented as the mean cell number with standard deviation (s.d.) from the midguts per genotype or conditions counted in all three independent experiments. No blinding experiment has been performed. The sample size used in present study is based on previous publications[10,29].

**Chromatin immunoprecipitation.** 120 ATF3::GFP adult midguts were dissected in ice-cold PBS and fixed in PBS containing 1% formaldehyde (Thermo scientific, formaldehyde ampules, methanol free) for 30 min. Enough glycine was added to a final concentration of 125 mM to stop the crosslinking. Samples were then spun down at 4 °C, 800g for 5 min and washed with ten times volume of ice-cold PBS plus protease inhibitors. Fixed tissues were homogenized in ice-cold PBS using rotor stator for 5 min and centrifuge down the homogenized sample at 4 °C, 800g for 5 min. The tissue pellet were resuspended in 2 ml sonication buffer (10 mM Tris pH8.0, 200 mM NaCl, 1 mM EDTA, 0.5% N-lauroylsarcosine, 0.1% Na-deoxycholate, protease inhibitor) and sonicated with a M220 Focused-ultrasonicator (Covaris, Woburn, Massachusetts, USA) for 30 min on ice-water bath. This resulted in 200–500-bp fragments. After centrifugation (4 °C, 15 min, 13,000g) and addition of Triton X-100 to a final concentration of 1%, the soluble chromatin was used directly for ChIP. Ten microlitres chromatin was precleared with 50 μl Protein A Sepharose beads. Two microlitres precleared chromatin was reserved as the input sample. Remaining chromatin was incubated with 0.2 mg antibody overnight at 4 °C. Antibody-protein complexes were incubated with 50 μl Protein A Sepharose beads and washed five times, 3 min each with LiCl buffer (250 mM LiCl, 10 mM Tris-HClpH 8.0, 1 mM EDTA, 0.5% NP-40, 0.5% sodium deoxycholate), followed by a TE with 50 mM NaCl wash. RNaseA and Proteinase K were added to remove the protein and RNA residues. The samples were precipitated with isopropanol and washed with 70% ethanol. After DNA purification, IP and input pellets were resuspended in H₂O. The suspension was used per qPCR reaction. For each locus three independent ChIP-qPCR experiments were performed. For ChIP sequencing, the DNA was prepared as described above except 150-200 midguts were dissected for each sample. DNA was retrieved and then cloned into a sequencing library and subjected to Illumina HiSeq2000 for sequencing with single reads of 50 bps length. Sequencing reads were aligned to the reference genome of *Drosophila melanogaster* (UCSC dm3) using Bowtie 0.12.7. The model-based analysis of ChIP-seq (MACS 1.4.1) were used to identify the regions of ATF3::GFP enrichment over input DNA. After peak calling, the enrichments were annotated and linked to gene with the Galaxy platform[70].

**Microscopy and image analysis.** Images of whole intestine tissue were acquired on a Leica SP5 confocal microscope using tile scan and a × 20 objective. Images of adult intestines with immunofluorescence staining were acquired using a Leica SP5 scanning confocal, with a × 63 oil objective. The laser intensity and

background filtering was always set according to the control samples and remained the same for all subsequent samples. The colour intensity of most images has been enhanced equally for all images within the same experiment using linear adjustments in Photoshop CS5. The quantification of pH3 staining was done by manually counting on the SP5 confocal microscope with required focal plate shift. The length of the whole intestine was analysed using NeuronJ, an ImageJ plugin to facilitate the tracing and quantification of the tissue in images. All images were processed with Adobe Photoshop and Illustrator.

**Statistical analysis.** For all experiments, statistical significance was calculated in GraphPad Prism 5 for Mac OS X using the Student's *t*-test (two-tailed, two sample equal variance). * equals $P < 0.05$, ** equals $P < 0.01$ and *** equals $P < 0.001$. All experimental groups within one experiment shown in the figure were statistically tested. For survival curves, the Log-rank test was used in this study.

**Data availability.** The data that support the findings of this study are available from the corresponding author on request.

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

## Acknowledgements

We thank A. Boettcher, V. Chaudhary, J. Grossa, J. Mattila and V. Li for helpful comments on the manuscript. We are grateful to M. Uhlirova, A. Goto, B. Lemaitre, N. Perrimon, FlyORF, VDRC, Bloomington Drosophila Stock Centers for reagents and fly stocks. We thank the FACS Sorting Core Facility (ZMBH) and the Genomics and Proteomics Core Facility of DKFZ for help with experiments, N. Ha and J. Mallm (BioQuant) for help with sequencing data analysis. K. Han, J. Maldera, C. Pallares-Cartes, J. Xiang and members of the Edgar and Boutros labs for discussions and comments. Research in the laboratories of M.B. and B.A.E. is supported in part by the DFG Collaborative Research Center SFB873. Research in the lab of B.A.E. was supported by ERC AdG 268515.

## Author contributions

J.Z. performed the experiments, analysed the data and wrote the manuscript. B.A.E. analysed the data and wrote the manuscript. M.B. conceived the study, analysed the data and wrote the manuscript.

## Additional information

**Competing financial interests:** The authors declare no competing financial interests.

