## [Peer Review File · Nature Communications]

Reviewers' comments:

Reviewer #1 (Remarks to the Author):

'ATF3 acts as a rheostat to control JNK signaling in intestinal regeneration and inflammation'

In this report, Jun Zhou et al., describe a new component in the machinery involved in the regulation of ISC proliferation in response to damage/stress to the fly midgut. They find that ATF3-Raw work to modulate JNK activation and apoptosis in the ECs during homeostasis and in response to tissue damage. This impacts JAK/Stat signaling activity and ISC proliferation.

This is a well-structured story with experiments carefully done and controlled for. The data presented and conclusions drawn from it are generally convincing. Proper quantifications and statistical analyses have been presented.

ATF3 has been previously assigned roles in JNK signaling and in the larval midgut. However, the authors present and characterize a novel role for this transcription factor in the adult midgut and also identify raw as a direct target of ATF3.

I suggest addressing of the following issues before consideration for publication:

- Have the authors tested the effect of knocking down ATF3 in the stem/progenitor cells (esgts>ATF3-IR)? Given its expression pattern, it is possible that ATF3 also plays a role in those cells.
- The levels of puc transcript presented in Figure 3(H-K) should be quantified by qRT-PCR.
- Genetic interactions similar to the ones done for ATF3 should be performed between Raw and the JAK/Stat pathway to further support the conclusions.
- The resistance to bacterial infection and reduced regeneration in midguts overexpressing ATF3 is interesting but rather unexplored. The authors should look at which if any of the multiple damage-responsive pathways (JNK, JAK/Stat, Wnt, Hippo, EGFR, etc) may be affected in ATF3 overexpressing guts subject to infection?
- The significance of the data presented in Figure 5F is unclear as the effects seen rather additive.
- The Caspase staining presented in Figure 6 are not very convincing. Should this not label cell nuclei? TUNEL staining like the ones presented in Suppl. Figure 6 are more convincing.
- How do the authors explain the rescue of Hep (Act) midguts by overexpressing ATF3 considering that ATF3 is upstream of the JNK pathway?
- What is the functional role of ATF3 in the ageing phenotypes of the midgut?
- Figure 6 appears labelled as 5 and Figure 5 has no label.

Reviewer #2 (Remarks to the Author):

This study from Zhou et al., analyze one role of ATF3 in gut homeostasis of *Drosophila*. The role of this transcription factor in the gut has been previously examined by the Uhlirova and the Lemaitre groups, however this study reports a new function of ATF3, which acts as a JNK antagonist via the regulation of *raw*. Overall this is an interesting manuscript that provides evidence that ATF3 plays essential roles in intestinal homeostasis by balancing the activation of the JNK dependant apoptosis and epithelium proliferation. Globally, the data provided in the paper are convincing, however, a number of point need to be addressed before this study can be published. One issue is that this paper gives the impression that most of function of ATF3 is mediated through *Raw*. I wonder if the authors could not use their Chip Seq approach to identify ATF3 target gene. In addition, the article needs some editorial attention as there are many instances where Figures and their legends are wrongly indicated and do not match to the text. There are similar concerns in regards to citation numbers being incorrect.

Major Concerns:

1. In Supplementary Figure 1b the fold induction of ATF3 with ageing is not quite convincing. The authors should assess the impact of ageing at later time points, such as 40 days and 60 days, as the medium lifespan of wild-type strains is approximately 45 days.
2. In a similar vein as the point above, the lifespan of *MyoIAts>ATF3-RNAi* should be determined to understand the impact of increased intestinal proliferation and subsequent dysplasia on the survival of flies lacking ATF3 in the gut.
3. The Uhlirova group has previously shown an increase in bacteria in the guts of *atf3*[76] mutant larvae. Can the increase in proliferation under basal conditions of the *MyoIAts>ATF3-RNAi* strain be rescued by feeding antibiotics or use of germ-free condition, i.e. does the microbiota impact this phenotype.
4. The authors keep switching between the adult gut specific driver *MyoIAts* and the ubiquitous driver *Actts* throughout the paper (e.g. Fig 2K&L + Fig 4I&J). This is quite confusing, as some phenotypes such as increase of 10X-STAT-GFP reporter in the gut is never demonstrated with a gut-specific knockdown of ATF3. Also in these experiments with the *Actts* driver the authors have not indicated whether they have used RNAi #1 or #2.
5. Is the impact on phosphorylation of JNK upon ATF3 cell-autonomous? The authors should monitor phospho-JNK in a clonal analysis using *esgts F/O* to knock down ATF3.
6. In Figure 2E and 2M the 'n' for PH3 counts for ATF3IR, *KayIR* and ATF3IR, *Upd3IR* is quite low. In the methods section for determining PH3 counts, the authors indicate that at least 8-10 guts were taken per experiment and this was repeated for 3 biological replicates. If this is the case, the indicated 'n' (n=11, n=14) does not match up in these Figures.
7. For the ChIP-seq data, it is not clear whether 2 biological repeats were taken to do this analysis from the methods. Also as the authors talk about the auto-regulation of ATF3 but do not show the binding peaks from the ChIP-Seq data this is hard to understand. It seems that this data will be part of another manuscript, but I think clearer validation of this data in this paper the authors need to show ChIP-PCR for other genes such as ATF3. They should also check nearby genes of *raw* that show smaller peaks CG9314 by ChIP-qPCR. Globally, the strenght of the paper will be reinforced if they could use their method to identify ATF3 target gene. One issue with the paper is that it seems that all the effect of ATF3 is mediated by *raw* while ATF3 could have multiple other role in intestinal homeostasis.
8. The cell-type specific expression of *raw* in the gut using the *raw-GFP* enhancer trap line (Figure 3D-F) should include co-staining for ISC marker *Delta* and EC marker *Pdm1*.

9. Armadillo staining in Supp Figure 4a should have insets with enlarged images as the current Figure it is difficult to see the dysplasia and cell boundaries properly. I am not sure whether a sign of dysplasia is in fact mislocalized Arm or increased Arm staining as stated by the authors.

10. The authors show that over-expression of raw in ATF3IR flies rescue the increase in stem cell proliferation. Does over-expression of raw have any impact on reducing PH3 counts after P.e infection?

11. In the discussion, the authors say that data not shown indicated that ATF3 deficiency leads to decrease in antimicrobial production. This statement contradicts Rynes et al, 2012 findings where their microarray and RT-qPCR data shows antimicrobial peptide AttD to be increased in the guts of the *atf3*[76] mutant animals. This data could be include in the paper and it might be interesting to know whether the impact of ATF3 on antimicrobial peptide genes expression is dependent of Raw.

Minor points

1. The age of flies should be indicated consistently in the Figure legends throughout the paper. For example, age of flies is missing from the RT-qPCR analysis done on genes of the JAK/STAT and JNK pathways in flies where ATF3 is knocked down in Supplementary Figure 2A.

2. Figure panels 2G and 2H should come below Fig 2I and 2J as these results is discussed first in the text and the Figure numbers should be in order of being mentioned in the text.

3. The source of the *Upd3-LacZ* and *puc-LacZ* lines is missing for the materials and methods, and the *Upd3-LacZ* line should be referenced in the manuscript on Page 7 as well.

4. *puc-LacZ* staining in *esgts F/O* is not Figure 2E as indicated in the legends but rather Figure 2I & 2J.

5. The induction of raw in the PGRP-LC mutant strain upon infection with P.e is not discussed in the manuscript for Figure 3C.

6. Figure 5 is not labeled and Figure 6 is mis-labelled as Figure 5.

7. On Page 9 of the text, the authors should cite where infection-induced intestinal dysplasia has been previously shown.

8. In Figure 5C-E the anterior part of the gut (proventriculus) is not visible in the images. Also in Supp Fig6 E-H the orientation of the gut is inverted with anterior side of the gut on the right instead of the left.

9. On page 10 the PGRP-LC mutant strain is not cited, from which study the authors have used this mutant. A list of all the mutants used could be useful.

10. On page 12 the coexpression of p35 along with ATF3 leads to reduction of proliferation. The authors should cite Domingo and Steller 2007 and Amcheslavsky et al 2009.

11. In the Discussion on Page 13 lines: 'We found that ATF3 depletion in the *Drosophila* intestine resulted in in appropriate JNK activation, and thereby increased epithelial cell death and stem cell activation for proliferation.' the authors should replace 'in appropriate' with inappropriate.

12. On Page 14 the raw embryogenesis paper is miscited as 51 that should be 41, and 57 for Byars et al and not 51.

13. Many genes, genotype and RNAi are not italicized.

Reviewer #3 (Remarks to the Author):

Different types of stress invoke a regenerative response in the *Drosophila* gut that entails proliferation and differentiation of intestinal stem cells. It had previously been shown that JNK signaling and activation of the leucine zipper transcription factor Fos can mediate this response. The paper submitted by Zhou et al. presents evidence for a role of a different, less well studied leucine zipper factor, ATF3, in the intestinal stress response of *Drosophila*. The data show convincingly that ATF3 expression is increased in response to oxidative stress, JNK signaling, or bacterial infection in enterocytes. There it antagonizes JNK signaling and serves to down regulate the JNK response. The mechanism by which ATF3 down-regulates JNK depends on the raw gene product, a protein of poorly characterized function that had previously been reported to function as a negative regulator of JNK signaling. Several lines relying on gain and loss of function experiments for Raw and ATF3 support the model of a negative feedback loop involving these factors. Loss of either ATF3 or raw causes an overshooting JNK response that results in excessive proliferation, apoptosis, loss of intestinal integrity and death.

The experimental data support the drawn conclusions. The novel information presented in this manuscript add another layer to our increasingly sophisticated understanding of the signaling network that regulates intestinal homeostasis and stem cell function in the *Drosophila* model. The new information on the function of ATF3 may shed light on the contribution of this factor to other biological processes. The paper should be of general interest to the readership of nature communications.

Minor comments:

Overall there are a few rough spots with the English, especially in the figure legends. Some careful copyediting would be advisable.

Specifically:

Figure 1 G & H: Does the green channel really show prospero? This looks like EC nuclei.

Fig 2 a and B: I assume the GFP comes from UAS GFP? That should be mentioned in the legend and/or by a label in the figure itself.

Fig 2 C and D: How informative is it that these guts have high levels of JNK activity. These are rare escapers and possibly quite sick and compromised. There are many reasons why they could be stressed in addition to a direct effect of ATF3 deficiency. An *atf3*^{-/-} clone would be more informative. However, this is not a critical experiment and can also be omitted.

Fig 3 J and K, and (especially) suppl. Figure 3 C and D: The suppression of *puc* LacZ expression by raw expression does not look very convincing. Can those data be quantified?

Numbering: Two figure 5's and no 6.

Reviewer #4 (Remarks to the Author):

Zhou and colleagues identify the transcription factor ATF3 as a key regulator of JNK signaling to maintain tissue homeostasis in the *Drosophila* intestine during bacterial infection. Using an ATF3::GFP reporter, the authors find that ATF3 is mainly expressed in Enterocytes (ECs) of the gut, and its expression is increased during ageing and in response to infection with P.e.. Knockdown of ATF3 in ECs leads to activation of JNK (assessed by *puc-lacZ*) and Jak/STAT signaling (based on *upd3-lacZ* and Stat::GFP), which then promotes intestinal stem cell (ISC) proliferation. To explore the mechanism by which ATF3 regulates JNK activity, the authors perform ChIP-seq and identify the gene *raw* as an ATF3 target, which is a known JNK antagonist. Loss of *raw* in ECs phenocopies the ISC hyper-proliferation induced by ATF3 deficiency. Genetic evidence suggests that *raw* acts downstream of ATF3 and upstream of JNK to control Jak/STAT signaling. The authors show that the ATF3-*raw* axis is required to limit JNK activity during P.e infection,

preventing excessive EC cell death, and promoting survival of the animal, likely due to improved barrier function.

The study provides interesting insight into how JNK activity is balanced during pathogen infection to avoid excessive cell death in the intestinal epithelium. Overall, the model and interpretations are well-supported and publication can be recommended. For the broad audience of Nature Communications, some more mechanistic insight into how ATF3 is regulated and how ATF3 regulates JNK would be useful:

1. the authors show that ATF3 expression increases in response to infection with P.e., but do not explore the mechanism of ATF3 regulation. It is shown that induction of ATF3 occurs both in response to infection and after Paraquat treatment, suggesting that its regulation is likely in response to epithelial damage. Likely candidates for regulators of ATF3 would thus be JNK itself or JAK/STAT signaling.

2. The regulation of JNK signaling is complex. The induction of puc in response to JNK activation sets up a negative feedback to limit JNK activity. It is not clear why this established negative feedback is not sufficient to limit JNK activity in response to infection, and how ATF3 play into this regulation. The authors could test whether ATF3 or Raw cooperate with puc to control JNK activity. In Fig. 4H it is shown that reducing one copy of puc results in additive increase in ISC proliferation, suggesting that ATF3/raw may regulate JNK signaling independently of puc. How does this interaction play out in infection experiments? How does it influence EC apoptosis? Can Puc over-expression rescue ATF3/raw loss of function phenotypes?

Response to reviewers' comments

We thank the reviewers for their helpful and generally positive comments. We have addressed the specific comments with additional experiments, as listed below.

Summary of changes:

1. New Fig. 2e and Supplementary Fig. 2e: Phospho-JNK staining in ATF3-RNAi esg F/O clones
2. New Fig. 3k: Quantification of puc-lacZ positive cells
3. New Fig. 3f: Induction of raw-GFP in ATF3 overexpressing midgut
4. New Fig. 4h-i: Induction of Upd3-lacZ was observed in *raw* depleted midgut
5. New Fig. 4l: Co-RNAi raw and Upd3 and performed pH3 quantification
6. Revised Fig. 2f, 2m, 4e and 5f: Performed additional experiments and added pH3 counts
7. New Supplementary Fig. 1i-n: RNAi ATF3 in intestinal EEs (*Pros^{ts}*) or VMs (*How^{ts}*) and performed PH3 staining
8. New Supplementary Fig. 3a-b: Binding peak on ATF3 gene and ChIP-qPCR validation
9. New Supplementary Fig. 3d-e: Co-staining of raw-GFP with Delta-lacZ or MyoIA-lacZ
10. New Supplementary Fig. 4b,d,f: Enlarged images of Armadillo staining
11. New Supplementary Fig. 4h-j: ATF3 and raw expression in JNK activation or suppression condition
12. New Supplementary Fig. 5b: RT-qPCR analysis of ATF3 overexpressing intestine

Reviewer #1

“In this report, Jun Zhou et al., describe a new component in the machinery involved in the regulation of ISC proliferation in response to damage/stress to the fly midgut. They find that ATF3-Raw work to modulate JNK activation and apoptosis in the ECs during homeostasis and in response to tissue damage. This impacts JAK/Stat signaling activity and ISC proliferation. This is a well-structured story with experiments carefully done and controlled for. The data presented and conclusions drawn from it are generally convincing. Proper quantifications and statistical analyses have been presented. ATF3 has been previously assigned roles in JNK signaling and in the larval midgut. However, the authors present and characterize a novel role for this transcription factor in the adult midgut and also identify raw as a direct target of ATF3.

I suggest addressing of the following issues before consideration for publication:”

“-Have the authors tested the effect of knocking down ATF3 in the stem/progenitor cells (esgts>ATF3-IR)? Given its expression pattern, it is possible that ATF3 also plays a role in those cells.”

This is a relevant question. Yes, we have tested ATF3 function in the ISCs by expressing RNAi using the *esg^{ts}* driver. We observed an induction of mitotic cells in the midgut of *esgts>ATF3-RNAi* flies. However, the *esgts>ATF3-RNAi* flies are not sensitive to *P.e* infection (data not shown). Since we did not observe high ATF3 expression in stem cells, we think the mitotic effect may be due to RNAi effect passing over to enterocytes since *esg* is a relative strong ISC/EB driver. In addition, we carefully examined the role of ATF3 in EEs and visceral muscle. We did not observe any effects on ISC mitosis following expression of ATF3 RNAi using *Pros^{ts}* or *How^{ts}* (New Supplementary Fig. 1i-n in the revised manuscript). Hence, we believe that the function of ATF3 in the ECs is most important for maintaining gut homeostasis and for the host defense under stress or infection condition. Consequently, this has been the focus of our manuscript.

“-The levels of puc transcript presented in Figure 3(H-K) should be quantified by qRT-PCR.”

Puckered expression is overall relatively abundant in the fly intestine. We believe that *puckered-lacZ* is more sensitive for detecting puckered expression and in addition provides good spatial resolution. In fact, many labs have noted that *puc-LacZ* is a more sensitive readout for stress response than puckered mRNA levels. Therefore, we have performed additional experiments and have in addition quantified the number of Puckered-lacZ positive cell per image per genotype. These improved data are shown in New Figure 3k of our revised manuscript.

“-Genetic interactions similar to the ones done for ATF3 should be performed between Raw and the JAK/Stat pathway to further support the conclusions. “

To answer the reviewer’s comment, we performed additional experiments by co-expression of RNAi targeting *raw* and *Upd3*, using *MyoIA^{ts}* driver. Consistent with our conclusions, we observed that *Upd3* knock down suppresses *raw* RNAi induced ISC mitosis (New Figure 4l in revised manuscript). In addition, EC-specific *raw* knock down induced *Upd3-lacZ* expression, which is consistent with our qPCR assay (Figure 4h-i in revised manuscript). Hence, we conclude that a signaling cascade of *ATF3*→*raw*→*JNK*→*JAK/STAT* occurs to control stem cell activity to maintain tissue homeostasis.

“-The resistance to bacterial infection and reduced regeneration in midguts overexpressing ATF3 is interesting but rather unexplored. The authors should look at which if any of the multiple damage-responsive pathways (JNK, JAK/Stat, Wnt, Hippo, EGFR, etc) may be affected in ATF3 overexpressing guts subject to infection?”

We tested the damage-response pathway activation in midguts by qPCR, and we observed that *Upd2*, *Upd3*, *Socs36E* (STAT), *Wg* (Wnt) and *Puckered* (JNK) levels are reduced in ATF3 overexpressing midguts (New Supplementary Fig. 5b in revised manuscript). These results suggest that ATF3 expression causes JNK suppression, which affects downstream signaling via STAT and Wnt.

“-The significance of the data presented in Figure 5F is unclear as the effects seen rather additive.”

To cite an example shown in Fig. 5f, 4 hrs *P.e* infected ATF3 RNAi midguts showed 112 ± 17 mitoses, which is higher than the combination of ATF3 RNAi (52 ± 12) and 4 hrs *P.e* infected WT (20 ± 12). Hence, ATF3 or raw RNAi induced proliferation is enhanced in response to infection, rather than an additive effect.

“- The Caspase staining presented in Figure 6 are not very convincing. Should this not label cell nuclei? Tunnel staining like the ones presented in Suppl. Figure 6 are more convincing. ”

We thank the reviewer for the comment. The staining by the anti-cleaved *Death Caspase 1* (DCP1) is cytoplasmic. Based on our experiences, TUNEL labels cell nuclei and is better suited for detecting massive apoptotic events. However, TUNEL staining gave strong background signals in some experiments with mild apoptotic phenotypes, for instance, *MyoIats>ATF3 RNAi, UAS-raw*. Instead, we used DCP1 staining for detecting apoptotic events in the intestine. Hence, we consider the DCP1 staining shown in the Fig. 6 would better represent the induced apoptosis in the intestine with various genotypes.

“-How do the authors explain the rescue of Hep (Act) midguts by overexpressing ATF3 considering that ATF3 is upstream of the JNK pathway?”

We believe that ATF3 functions downstream of JNKK (Hemipterous, Hep) and directly on (upstream of) JNK (Basket, Bsk) to control JNK signaling activity. We have shown two lines of evidence to support our hypothesis: First, Hep activation (Hep^{Act}) -induced mitosis, cell death and lethality can be rescued by ATF3 expression (Fig. 6g-k and Supplementary Fig. 6i). Second, expressing a dominant negative form of Basket (JNK) or Kay RNAi suppresses ATF3 RNAi induced stem cell mitosis (Fig. 2f). We have clarified this in the Results of revised manuscript by adding a sentence “these data indicate that ATF3 and raw act downstream of Hemipterous (JNKK) and upstream of Basket (JNK) to control JNK mediated regeneration and cell death” (Line 24-26 of page 12 in the revised manuscript). To further address whether ATF3-raw-JNK function as a feedback loop to auto-regulate JNK activity, we performed additional experiments to examine the ATF3 and raw mRNA levels in JNK activation and inactivation conditions (New Supplementary Fig. 4i-j). We found that JNK activation induces ATF3 and raw expression in the intestine. However, JNK suppression did not cause reduction in ATF3 and raw expression in presence or absence of infection (New Supplementary Fig. 4i-j). Taken together, these observations suggest that ATF3-raw regulates the JNK pathway through Basket, and that the expression of ATF3 and raw can be regulated by gut stress independently of JNK.

“-What is the functional role of ATF3 in the ageing phenotypes of the midgut?”

We have not investigated the function of ATF3 during ageing in detail. A possible function of ATF3 could be that it restricts JNK / inflammatory signaling during ageing, however, further studies will be needed. We have discussed this in the discussion: “In addition, JNK signaling has deleterious effects on longevity and is activated during ageing for the proliferative homeostasis, whereas limiting JNK activation extends lifespan (Biteau et al., 2010). As ATF3 regulates raw expression during ageing, it would be also interesting to explore the roles of ATF3 and raw in regulating JNK signaling during the ageing processes.” (Line 13-17 of page 16 in the revised manuscript).

“- Figure 6 appears labelled as 5 and Figure 5 has no label.”

Thank you. We have corrected this.

Reviewer #2

This study from Zhou et al., analyze one role of ATF3 in gut homeostasis of Drosophila. The role of this transcription factor in the gut has been previously examined by the Uhlirova and the Lemaitre groups, however this study reports a new function of ATF3, which acts as a JNK antagonist via the regulation of raw. Overall this is an interesting manuscript that provides evidence that ATF3 plays essential roles in intestinal homeostasis by balancing the activation of the JNK dependant apoptosis and epithelium proliferation. Globally, the data provided in the paper are convicing, however, a number of point need to be addressed before this study can be published. One issue is that this paper gives the impression that most of function of ATF3 is mediated through Raw. I wonder if the authors could not use their Chip Seq approach to identify ATF3 target gene. In addition, the article needs some editorial attention as there are many instances where Figures and their legends are wrongly indicated and do not match to the text. There are similar concerns in regards to citation numbers being incorrect.

We thank the reviewer for the positive comments on our manuscript. We agree that ATF3 plays a complex role in different biological processes, such as stress response, development, and metabolism. Our manuscript identifies an essential role of raw in the context of acute infection and shows that it is a transcriptional target of ATF3 and regulates JNK signaling in the intestine. In the revised manuscript, we also show that ATF3 binds its own promoter to regulate its own expression (New Supplementary Fig. 3a-b). Moreover, we found that Lipase 4 is one of ATF3 targets in intestine (data not shown), which is consistent with the role of ATF3 in regulating lipid homeostasis (Rynes et al., 2012; Chakrabarti et al., 2014). However, the detailed mechanism of other potential transcriptional targets of ATF3 in different conditions, such as infection, starvation, development or ageing still need to be further explored and cannot be included here. We thank the reviewer for pointing out several errors, which we have corrected in our revised manuscript.

“Major Concerns:

1. In Supplementary Figure 1b the fold induction of ATF3 with ageing is not quite convincing. The authors should assess the impact of ageing at later time points, such as 40 days and 60 days, as the median lifespan of wild-type strains is approximately 45 days.”

In our RNAi experiment, the flies were kept at 29 °C and the median life span of wildtype flies is around 35 days. A previous study identified PGRP-SC2 as a longevity promoting factor in *Drosophila* ageing intestine using a genome wide transcriptome analysis. From their RNAseq data analysis, we found ATF3 expression is also increased during ageing (0, 2, 10, 20, 30, 40 days) (Guo et al., 2014).

“2. In a similar vein as the point above, the lifespan of MyoIAts>ATF3-RNAi should be determined to understand the impact of increased intestinal proliferation and subsequent dysplasia on the survival of flies lacking ATF3 in the gut.”

We would thank the reviewer for this interesting suggestion. We also think exploring the role of ATF3 in ageing is a very interesting question and we would like to explore this in the future, however, given the relatively long times required for such experiments, we were unable to provide them in this revision.

“3. The Uhlirova group has previously shown an increase in bacteria in the guts of atf3[76] mutant larvae. Can the increase in proliferation under basal conditions of the MyoIAts>ATF3-RNAi strain be rescued by feeding antibiotics or use of germ-free condition, i.e. does the microbiota impact this phenotype.”

This is an interesting question. We would like to point out ATF3 RNAi can induce JNK activity and stem cell proliferation in a very short time, for example 24 hours. We do not believe intestinal microbiota can be significantly expanded in such a short time point to result in strong cell death and stem cell proliferation, except in cases when flies are fed with high dose of bacteria such as *Pseudomonas entomophila* (*P.e*), *Erwinia carotovora carotovora* 15 (*Ecc15*) and others. In addition, the RNAseq analysis from Guo et al., (2014) also reveals ATF3 is increased in ageing intestine in axenic condition, which suggest ATF3 is increased in response to ageing independently of microbiota.

“4. The authors keep switching between the adult gut specific driver MyoIAts and the ubiquitous driver Actts throughout the paper (e.g. Fig 2K&L + Fig 4I&J). This is quite confusing, as some phenotypes such as increase of 10X-STAT-GFP reporter in the gut is never demonstrated with a gut-specific knockdown of ATF3.

The experiment cannot be done with the available enterocyte specific driver-*MyoIA^{ts}* (Jiang et al., 2009) because this driver also has UAS-GFP transgene and the GFP signal is seen throughout the whole gut. For this reason, the STAT-GFP reporter could not be monitored, and so we used *Act^{ts}* instead of *MyoIA^{ts}*. Although we agree that this is somewhat inelegant, it was a practical solution and we believe the results are solid and relevant.

Also in these experiments with the Actts driver the authors have not indicated whether they have used RNAi #1 or #2.”

In the revised manuscript, we have clearly indicated which ATF3 RNAi was used in all ATF3 RNAi related experiments.

“5. Is the impact on phosphorylation of JNK upon ATF3 cell-autonomous? The authors should monitor phospho-JNK in a clonal analysis using esgts F/O to knock down ATF3.”

We thank the reviewer for pointing this out. We performed the suggested experiment of phospho-JNK staining on the intestine of *esgF/O>ATF3 RNAi* flies. As shown in New Figure 2e and New Supplementary Figure 2e in the revised manuscript, the phospho-JNK signal is only detectable in ATF3 RNAi flipout clones, but not in the neighboring wild type cells. This indicates a cell autonomous effect and is consistent with the puckered-lacZ reporter assay (revised Supplementary Figure 2b).

“6. In Figure 2E and 2M the 'n' for PH3 counts for ATF3IR, KayIR and ATF3IR, Upd3IR is quite low. In the methods section for determining PH3 counts, the authors indicate that at least 8-10 guts were taken per experiment and this was repeated for 3 biological replicates. If this is the case, the indicated 'n' (n=11, n=14) does not match up in these Figures.”

We could not count all the guts in the experiment as some guts that were ‘stuffed’ with food had internal gut florescence or were damaged during the staining procedure. Therefore, not every genotype had over 20 guts counted in the experiment. We have repeated several experiments to add more pH3 counts per genotype per experiment as shown in Fig. 2f, Fig. 2m, Fig. 4e and Fig. 5f in the revised manuscript.

7. For the ChIP-seq data, it is not clear whether 2 biological repeats were taken to do this analysis from the methods. Also as the authors talk about the auto-regulation of ATF3 but do not show the binding peaks from the ChIP-Seq data this is hard to understand. It seems that this data will be part of another manuscript, but I think clearer validation of this data in this paper the authors need to show ChIP-PCR for other genes such as ATF3. They should also check nearby genes of raw that show smaller peaks CG9314 by ChIP-qPCR. Globally, the strength of the paper will be reinforced if they could use their method to identify ATF3 target gene. One issue with the paper is that it seems that all the effect of ATF3 is mediated by raw while ATF3 could have multiple other role in intestinal homeostasis.

We thank the reviewer for the comments. For the present manuscript, we did not perform a full genomic and functional analysis of ATF3 target genes but focused on selected genes that show a clear phenotype. We indeed found that ATF3 can bind its own promoter and ChIP-qPCR confirmed the binding (New Supplemental Fig. 3a in revised manuscript). In revised Fig. 3a, the smaller peaks are on the exon of CG9314 gene locus, which we would not consider as binding peak but rather noise peak. We also found other

targets of ATF3 like Lipase 4, which is consistent with the finding from Chakrabarti et al., (2014), and their study showed ATF3 is involved in intestinal lipid metabolism. We believe ATF3 has multiple targets which exact detail has to be further explored. It has been reported that raw negatively regulates JNK pathway during embryogenesis (Bates et al., 2008). We found ATF3 influences tissue regeneration through JNK activation. Therefore, it is reasonable to select raw as one of the ATF3 targets and involved in the regulation of JNK signaling during infection. In this manuscript, we provide multiple lines of evidence that raw is a key target of ATF3 and is involved in the regulation of JNK mediated stresses response. A full genomic analysis of ATF3's targets under multiple conditions is in our view beyond the scope of our study.

“8. The cell-type specific expression of raw in the gut using the raw-GFP enhancer trap line (Figure 3D-F) should include co-staining for ISC marker Delta and EC marker Pdm1. “

We performed additional experiments to confirm raw-GFP is exclusively expressed in enterocytes by combining raw-GFP line with either stem cell (*Delta-lacZ*) or enterocyte reporters (*MyoIA-lacZ*). As shown in New Supplementary Figure 3d, e and revised Figure 3d in the revised manuscript, raw-GFP is expressed in *MyoIA-lacZ* positive cells, but not with *Delta-lacZ* or *propero* positive cells. These results indicate raw-GFP is specifically expressed in the enterocytes.

“9. Armadillo staining in Supp Figure 4a should have insets with enlarged images as the current Figure it is difficult to see the dysplasia and cell boundaries properly. I am not sure whether a sign of dysplasia is in fact mislocalized Arm or increased Arm staining as stated by the authors.”

We have added the enlarged images in New Supplementary Figure 4b, d, f in the revised manuscript. As described in the paper from Apidianakis et al., (2009), infection induces intestinal dysplasia. ATF3 and raw RNAi show similar defects such as mislocalized and increased Arm staining, as shown in Supplementary Figure 4a-f in the revised manuscript. In addition, we found that ATF3 RNAi synergizes with infection to induce stem cell proliferation (revised Figure 5f). Hence, we consider ATF3 or raw RNAi to show similar dysplasia phenotype as infection.

“10. The authors show that over-expression of raw in ATF3IR flies rescue the increase in stem cell proliferation. Does over-expression of raw have any impact on reducing PH3 counts after P.e infection?”

Yes, it does. As shown in revised Figure 5g-m, raw or ATF3 overexpression prevents *P.e* infection induced stem cell proliferation.

“11. In the discussion, the authors say that data not shown indicated that ATF3 deficiency leads to decrease in antimicrobial production. This statement contradicts Rynes et al, 2012 findings where their microarray and RT-qPCR data shows antimicrobial peptide AttD to be increased in the guts of the atf3[76] mutant animals.

This data could be include in the paper and it might be interesting to know whether the impact of ATF3 on antimicrobial peptide genes expression is dependent of Raw.”

Thank you for pointing this out. We think the difference between the previous finding and our study maybe due to two reasons. First, Rynes et al. looked at *atf3*⁷⁶ in larvae tissue whereas we test *MyoIAs>ATF3 RNAi* in adult midgut. The regulation of antimicrobial peptide expression by ATF3 may be indirect and distinct from larvae and adult *Drosophila*. The second reason is that the mutant larvae they were using in their study were pretty sick. The induction of antimicrobial peptides may be due to the increase bacteria in the unhealthy larvae mutant intestine; a very indirect effect.

We have performed conditional knock down of ATF3 RNAi using temperature sensitive system *MyoIA^{ts}* (*MyoIA-Gal4,UAS-GFP; TubGal80*), which we can exclude ATF3 loss of function cause the developmental defect effect. We found multiple antimicrobial genes to be reduced in ATF3 RNAi condition but induced in ATF3 overexpression condition (Reviewer Figure 1 as shown below). In addition, we generated ATF3 RNAi and overexpression flipout clones and examined Diptericin (Dpt) activity. We confirmed that ATF3-deficient cells show reduced Dpt expression whereas ATF3 overexpression induces Dpt expression in the clone cells (Reviewer Figure 1 as shown below). The properly balanced immune response is important for the animal's resistance to bacterial infection, whereas immune deficient animals (*Dredd^{B118}*, *Key¹* and *Relish^{E20}*) are highly susceptible to bacterial infection (Ryu et al., 2006). However, hyperactivation of Relish in flies by mutation of *Jra*, *Stat92E* or *Dsp1* also showed a reduced survival against infection (Kim et al., 2007). Meanwhile, Relish hyperactivated flies by lacking multiple negative regulators showed a reduced lifespan (Paredes et al., 2011). As shown in Figure 5a, ATF3 overexpression results in a better survival against infection, which is probably due to reduced JNK activity caused apoptosis suppression but not Relish hyperactivation. In addition, ATF3 RNAi caused susceptibility can be rescued by overexpression of *raw* or a dominant negative form of *basket* (Fig. 5b and Supplementary Fig. 7b), suggesting that JNK hyperactivation is the main reason that ATF3 RNAi flies are more sensitive to infection. Hence, we conclude ATF3-*raw*-JNK but not ATF3-*Imd/Relish* is primarily responsible for the ATF3 depletion-induced sensitivity to infection.

“Minor points

1. The age of flies should be indicated consistently in the Figure legends throughout the paper. For example, age of flies is missing from the RT-qPCR analysis done on genes of the JAK/STAT and JNK pathways in flies where ATF3 is knocked down in Supplementary Figure 2A.”

We have modified the revised manuscript accordingly.

“2. Figure panels 2G and 2H should come below Fig 2I and 2J as these results is discussed first in the text and the Figure numbers should be in order of being mentioned in the text.”

We have changed the order in the revised manuscript.

“3. The source of the Upd3-LacZ and puc-LacZ lines is missing for the materials and methods, and the Upd3-LacZ line should be referenced in the manuscript on Page 7 as well.”

Upd3-lacZ line was generated by the Edgar lab (Jiang et al., 2009). We have added the information of Puc-lacZ in material and methods. We have referenced the lines in the revised manuscript.

“4. puc-LacZ staining in esgts F/O is not Figure 2E as indicated in the legends but rather Figure 2I & 2J.”

We have changed this in the revised manuscript.

“5. The induction of raw in the PGRP-LC mutant strain upon infection with P.e is not discussed in the manuscript for Figure 3C.”

We have discussed the mild reduction of raw in the PGRP-LC mutant in the revised manuscript (Line 26-31 on page 9 in the revised manuscript).

“6. Figure 5 is not labeled and Figure 6 is mis-labelled as Figure 5.”

We have corrected the Figure labels in the revised manuscript.

“7. On Page 9 of the text, the authors should cite where infection-induced intestinal dysplasia has been previously shown.”

We have cited infection-induced intestinal dysplasia part in the revised manuscript.

“8. In Figure 5C-E the anterior part of the gut (proventriculus) is not visible in the images. Also in Supp Fig6 E-H the orientation of the gut is inverted with anterior side of the gut on the right instead of the left.”

We have changed the representative images of Fig. 5c-e in the revised manuscript. We have also changed the orientation of the tissue in Supplementary Fig. 6e-h in the revised manuscript.

“9. On page 10 the PGRP-LC mutant strain is not cited, from which study the authors have used this mutant. A list of all the mutants used could be useful.”

The PGRP-LC⁷⁴⁵⁷ mutant strain is a kind gift from Akira Goto, which has not been published. We have added the information of all the mutants used in the New Supplementary table 2.

“10. On page 12 the coexpression of p35 along with ATF3 leads to reduction of

proliferation. The authors should cite Domingo and Steller 2007 and Amcheslavsky et al 2009.”

We have cited these two papers in the part reviewer suggested in the revised manuscript (Line 28-32 on page 12).

“11. In the Discussion on Page 13 lines: ‘We found that ATF3 depletion in the Drosophila intestine resulted in in appropriate JNK activation, and thereby increased epithelial cell death and stem cell activation for proliferation.’ The authors should replace ‘in appropriate’ with inappropriate.”

We have changed this in the revised manuscript.

“12. On Page 14 the raw embryogenesis paper is miscited as 51 that should be 41, and 57 for Byars et al and not 51.”

We have changed this in the revised manuscript.

“13. Many genes, genotype and RNAi are not italicized.”

We have changed this in the revised manuscript.

Reviewer #3

“Different types of stress invoke a regenerative response in the Drosophila gut that entails proliferation and differentiation of intestinal stem cells. It had previously been show that JNK signaling and activation of the leucine zipper transcription factor Fos can mediate this response. The paper submitted by Zhou et al. presents evidence for a role of a different, less well studied leucine zipper factor, ATF3, in the intestinal stress response of Drosophila. The data show convincingly that ATF3 expression is increased in response to oxidative stress, JNK signaling, or bacterial infection in enterocytes. There it antagonizes JNK signaling and serves to down regulate the JNK response. The mechanism by which ATF3 down-regulates JNK depends on the raw gene product, a protein of poorly characterized function that had previously been reported to function as a negative regulator of JNK signaling. Several lines relying on gain and loss of function experiments for Raw and ATF3 support the model of a negative feedback loop involving these factors. Loss of either ATF3 or raw causes an overshooting JNK response that results in excessive proliferation, apoptosis, loss of intestinal integrity and death.

The experimental data support the drawn conclusions. The novel information presented in this manuscript add another layer to our increasingly sophisticated undstanding of the signaling network that regulates intestinal homeostasis and stem cell function in the Drosophila model. The new information on the function of ATF3 may shed light on the contribution of this factor to other biological processes. The paper should be of general interest to the readership of nature communications.”

“Minor comments:

Overall there are a few rough spots with the English, especially in the figure legends. Some careful copyediting would be advisable.”

Thank you. We have carefully revised the manuscript and paid particular attention to the Figure legends.

“Specifically:

Figure 1 G & H: Does the green channel really show prospero? This looks like EC nuclei.”

Thank you. It should read GFP instead of prospero. We have corrected this in the revised manuscript.

“Fig 2 a and B: I assume the GFP comes from UAS GFP? That should be mentioned in the legend and/or by a label in the figure itself.”

We have added this information in the Figure legends of the revised manuscript.

“Fig 2 C and D: How informative is it that these guts have high levels of JNK activity. These are rare escapers and possible quite sick and compromised. There are many reasons why they could be stressed in addition to a direct effect of ATF3 deficiency. An atf3 -/- clone would be more informative. However, this is not a critical experiment and can also be omitted.”

We appreciate the comment of the reviewer. The ATF3 hemizygous mutant are viable but sickly, however, we would like to point out that it is difficult to judge what is cause and consequence. We have added a sentence in the results to make this limitation transparent (Line 5-8 on page 6 in the revised manuscript).

“Fig 3 J and K, and (especially) suppl. Figure 3 C and D: The suppression of puc LacZ expression by raw expression does not look very convincing. Can those data be quantified?”

We thank the reviewer for this suggestion. We have quantified the Puc-lacZ positive cells in the revised manuscript and added it to New Figure 3k

“Numbering: Two figure 5's and no 6.”

Answer: Thank you, we have corrected this mistake.

Reviewer #4

“Zhou and colleagues identify the transcription factor ATF3 as a key regulator of JNK

signaling to maintain tissue homeostasis in the Drosophila intestine during bacterial infection. Using an ATF3::GFP reporter, the authors find that ATF3 is mainly expressed in Enterocytes (ECs) of the gut, and its expression is increased during ageing and in response to infection with P.e.. Knockdown of ATF3 in ECs leads to activation of JNK (assessed by puc-lacZ) and Jak/STAT signaling (based on upd3-lacZ and Stat::GFP), which then promotes intestinal stem cell (ISC) proliferation. To explore the mechanism by which ATF3 regulates JNK activity, the authors perform ChIP-seq and identify the gene raw as an ATF3 target, which is a known JNK antagonist. Loss of raw in ECs phenocopies the ISC hyper-proliferation induced by ATF3 deficiency. Genetic evidence suggests that raw acts downstream of ATF3 and upstream of JNK to control Jak/STAT signaling. The authors show that the ATF3-raw axis is required to limit JNK activity during P.e infection, preventing excessive EC cell death, and promoting survival of the animal, likely due to improved barrier function.

The study provides interesting insight into how JNK activity is balanced during pathogen infection to avoid excessive cell death in the intestinal epithelium. Overall, the model and interpretations are well-supported and publication can be recommended. For the broad audience of Nature Communications, some more mechanistic insight into how ATF3 is regulated and how ATF3 regulates JNK would be useful.”

“1. the authors show that ATF3 expression increases in response to infection with P.e., but do not explore the mechanism of ATF3 regulation. It is shown that induction of ATF3 occurs both in response to infection and after Paraquat treatment, suggesting that its regulation is likely in response to epithelial damage. Likely candidates for regulators of ATF3 would thus be JNK itself or JAK/STAT signaling.”

We thank the reviewer for the comments. As suggested, we tested ATF3, raw and JNK target gene expression under JNK suppression conditions (*MyoIats>UAS-Bsk^{DN}*) during infection. We did not see ATF3 or raw expression to be reduced upon JNK suppression (New Supplementary Figure 4j-l in revised manuscript), suggesting that JNK is not required for controlling ATF3 or raw expression. This is clearly stated in the revision. As JAK/STAT signaling is mainly active in progenitor cells and visceral muscle as indicated by the 10X STAT-GFP reporter, we assume that JAK/STAT signaling is not upstream signal on the regulation of ATF3 expression in enterocytes during infection. However, we observed that the p38c stress-response pathway regulates ATF3 expression in intestine (Reviewer Figure 2a is shown below), which is consistent with a previous finding (Chakrabarti et al., 2014).

“2. The regulation of JNK signaling is complex. The induction of puc in response to JNK activation sets up a negative feedback to limit JNK activity. It is not clear why this established negative feedback is not sufficient to limit JNK activity in response to infection, and how ATF3 play into this regulation. The authors could test whether ATF3 or Raw cooperate with puc to control JNK activity. In Fig. 4H it is shown that reducing one copy of puc results in additive increase in ISC proliferation, suggesting that ATF3/raw may regulate JNK signaling indepently of puc. How does this interaction play out in infection experiments? How does it influence EC apoptosis? Can Puc over-

expression rescue ATF3/raw loss of function phenotypes?”

We fully agree with the reviewer that the regulation of JNK signaling is tightly controlled and quite complex. In *Drosophila*, there is quite a lot of evidence that a number of additional negative regulators constrain JNK activity during development, such as Scarface, smt3, Parkin, Capping protein beta, falafel and others (Cha et al., 2005; Rousset et al., 2010; Huang et al., 2011; Fernández et al., 2014; Huang and Xue, 2015). Therefore, it is experimentally quite complicated to test whether the ATF3-raw module functions in parallel with other negative regulators to inhibit JNK activity in various conditions. As the reviewer mentioned, we have seen the synergistic effect of ATF3-raw and Puckered, suggesting both of them function to restrict JNK redundantly. However, we did not see that loss of one copy of Puckered enhances ATF3 depletion induced susceptibility to *P.e infection* (data not shown). We believe that the other JNK regulators play essential roles in controlling signaling activity in a context dependent manner.

However we did perform one experiment as the reviewer suggested, namely overexpressing Puc in the ATF3 RNAi condition. In this case we did not see that Puckered overexpression suppressed ATF3 RNAi-induced stem cell proliferation (Reviewer Figure 2b, shown below). This result indicates that Puckered functions independently of ATF3-raw in controlling JNK activity.

References

- Amcheslavsky, A., Jiang, J., Ip, Y.T. Tissue damage-induced intestinal stem cell division in *Drosophila*. *Cell Stem Cell*. 4: 49-61 (2009).
- Apidianakis, Y., Pitsouli, C., Perrimon, N. & Rahme, L. Synergy between bacterial infection and genetic predisposition in intestinal dysplasia. *P Natl Acad Sci USA* 106, 20883-20888. (2009).
- Bates, K.L., Higley, M. & Letsou, A. Raw mediates antagonism of AP-1 activity in *Drosophila*. *Genetics* 178, 1989-2002 (2008).
- Cha, G.H., Kim, S., Park, J., Lee, E., Kim, M., Lee, S.B., Kim, J.M., Chung, J., Cho, K.S. Parkin negatively regulates JNK pathway in the dopaminergic neurons of *Drosophila*. *Proc Natl Acad Sci U S A*. 102: 10345-50 (2005).
- Chakrabarti, S., Liehl, P., Buchon, N. & Lemaitre, B. Infection-induced host translational blockage inhibits immune responses and epithelial renewal in the *Drosophila* gut. *Cell host & microbe* 12, 60-70 (2012).
- Domingos P.M. & Steller, H. Pathways regulating apoptosis during patterning and development. *Curr Opin Genet Dev*. 17: 294-9 (2007)
- Fernández, B.G., Jezowska, B., Janody, F. *Drosophila* actin-Capping Protein limits JNK activation by the Src proto-oncogene. *Oncogene*. 33:2027-39 (2014).
- Guo, L., Karpac, J., Tran, S.L., Jasper, H. PGRP-SC2 promotes gut immune homeostasis to limit commensal dysbiosis and extend lifespan. *Cell* 156,109-22 (2014).
- Huang,H., Du, G., Chen, H., Liang, X., Li, C., Zhu, N., Xue, L., Ma, J., Jiao, R. *Drosophila* Smt3 negatively regulates JNK signaling through sequestering Hipk in the nucleus. *Development*. 138: 2477-85 (2011).

Huang, J.H. & Xue, L. Loss of flfl Triggers JNK-Dependent Cell Death in *Drosophila*. *BioMed Res Inter* 623573 (2015).

Jiang, H.Q., Patel, P.H., Kohlmaier, A., Grenley, M.O., McEwen, D.G. & Edgar, B.A. Cytokine/Jak/Stat Signaling Mediates Regeneration and Homeostasis in the *Drosophila* Midgut. *Cell* 137, 1343-1355 (2009).

Kim, L.K., Choi, U.Y., Cho, H.S., Lee, J.S., Lee, W.B., Kim, J., Jeong, K., Shim, J., Kim-Ha, J., Kim, Y.J. Down-regulation of NF-kappaB target genes by the AP-1 and STAT complex during the innate immune response in *Drosophila*. *PLoS Biol* 9: e238 (2007).

Paredes, J.C., Welchman, D.P., Poidevin, M., Lemaitre, B. Negative regulation by amidase PGRPs shapes the *Drosophila* antibacterial response and protects the fly from innocuous infection. *Immunity*. 35: 770-9 (2011).

Rousset, R, Bono-Laurio, S., Gettings, M., Suzanne, M., Spéder, P., Noselli, S. The *Drosophila* serine protease homologue Scarface regulates JNK signalling in a negative-feedback loop during epithelial morphogenesis. *Development*. 137: 2177-86 (2010).

Rynes, J. et al. Activating Transcription Factor 3 Regulates Immune and Metabolic Homeostasis. *Mol Cell Biol* 32, 3949-3962 (2012).

Ryu, J.H., Ha, E.M., Oh, C.T., Seol, J.H., Brey, P.T., Jin, I., Lee, D.G., Kim, J., Lee, D., Lee, W.J. An essential complementary role of NF-kappaB pathway to microbicidal oxidants in *Drosophila* gut immunity. *EMBO J* 25, 3693-701(2006)

Editorial Note: Figure R1 has been redacted to maintain the confidentiality of unpublished data.

Figure R2 p38c pathway regulates ATF3 expression in the intestine.

(A) The relative expression of *ATF3* mRNA in the midgut of control (*MyoIA^{ts}>w1118*), p38c overexpression (*MyoIA^{ts}>UAS-p38c*), p38c^{7B1/+} and p38c^{7B1} mutant flies, assayed by qRT-PCR. (B) Quantification of pH3-positive cells per adult midgut of the indicated genotypes after 5 days at 29°C.

REVIEWERS' COMMENTS:

Reviewer #1 (Remarks to the Author):

In this revised manuscript, Zhou and colleagues have addressed all my points well. I think this paper is now acceptable and will represent an important paper in the field.

Reviewer #2 (Remarks to the Author):

The authors have adequately answer to my request. I recommend the publication of the manuscript;

Below are two minor suggestions (to be taken or no in consideration by the author).

1) While the regulation of JNK pathway activation by ATF3-Raw is now adequately established, the claim that ATF3-Raw axis acts downstream of JNKK (Hep) and upstream of JNK (Bsk) is only supported with preliminary data. Over-expressing the strong gain-of-function version of JNKK (hepACT) is known to induce massive cell death (e.g. see Uhlirova&Bohmann 2006), compared to UAS-hepwt which induces relatively physiological levels of JNK activity. Therefore, use of the hepACT allele likely causes non-physiological effects beyond JNK activation. Another newly added result, that the MAP kinase phosphatase (puc) overexpression did not suppress ATF3-RNAi-induced stem cell proliferation, is confusing and does not fit with the final model that ATF3-Raw functions through JNK (bsk), since Puc can directly act on and inactivate JNK (Bsk). Moreover, placing ATF3-Raw between JNKK and JNK needs more stringent epistatic test and possibly molecular data to be solid. While I understand this should require many experiments therefore cannot be fully addressed in this manuscript, the authors should at least tone down their language and properly modify their final model.

2) I would suggest the authors remove the repression of ATF3 by itself, presented in Fig 7D "Predicted model for ATF3-raw regulates JNK activity". They do have data showing ATF3 binds to its own genomic locus, but this does not tell if those binding sites are required for activation or repression.

Reviewer #3 (Remarks to the Author):

The authors have addressed my concerns satisfactorily. However, there is a mistake in the labeling of the new panel K in figure 3. Raw over expression not Raw RNAi should suppress Puc LacZ. There was no experiment in which both ATF3 and raw RNAis are expressed in the corresponding panels g-j.

Reviewer #4 (Remarks to the Author):

The authors have responded appropriately to my original comments, performing additional experiments that point to a role for p38 MAPKinase in the control of ATF3 expression, and showing that Puc over-expression is not sufficient to inhibit ISC proliferation induced by the loss of ATF3. The authors only include these findings as additional reviewer data in the rebuttal letter, but they add interesting additional insight into the regulation of JNK signaling by ATF3 and Raw, and should, for this reason, be included in the main manuscript as supplemental data.

Response to reviewers' comments

We have addressed minor points as discussed below:

Summary of changes:

1. New Supplemental Fig. 4h: Inducing puc expression in ATF3 RNAi in enterocytes and performed PH3 staining
2. Fig. 6l, 7d and Supplemental Fig. 8: Modification of the model of ATF3-Raw controls JNK pathway activity.
3. We changed 'raw' to 'Raw' throughout the manuscript and figures.

REVIEWERS' COMMENTS:

Reviewer #1:

"In this revised manuscript, Zhou and colleagues have addressed all my points well. I think this paper is now acceptable and will represent an important paper in the field."

Reviewer #2:

"The authors have adequately answer to my request. I recommend the publication of the manuscript; Below are two minor suggestions (to be taken or no in consideration by the author).

1) While the regulation of JNK pathway activation by ATF3-Raw is now adequately established, the claim that ATF3-Raw axis acts downstream of JNKK (Hep) and upstream of JNK (Bsk) is only supported with preliminary data. Over-expressing the strong gain-of-function version of JNKK (hepACT) is known to induce massive cell death (e.g. see Uhlirova&Bohmann 2006), compared to UAS-hepwt which induces relatively physiological levels of JNK activity. Therefore, use of the hepACT allele likely causes non-physiological effects beyond JNK activation. Another newly added result, that the MAP kinase phosphatase (puc) overexpression did not suppress ATF3-RNAi-induced stem cell proliferation, is confusing and does not fit with the final model that ATF3-Raw functions through JNK (bsk), since Puc can directly act on and inactivate JNK (Bsk). Moreover, placing ATF3-Raw between JNKK and JNK needs more stringent epistatic test and possibly molecular data to be solid. While I understand this should require many experiments therefore cannot be fully addressed in this manuscript, the authors should at least tone down their language and properly modify their final model."

We would like to thank the reviewer for his suggestions and interpretation. We agree with the reviewer that Hep^{Act} leads to strong JNK activation and probably causes other side effects. We also would like to point out that *P.e* infection also activates multiple pathways (JNK, p38c, Duox, BMP, STAT, EGFR...) involved in a range of biological processes including stress response, immune response, tissue repair and

regeneration. We believe that additional input signals or molecules have interactive or synergistic effect on regulating JNK signaling in the tissue. Therefore, we believe we did not observe a complete rescue of ATF3 overexpression on tissue regeneration in either Hep^{Act} expression or *P.e* infection condition. Reducing one copy of *puc* enhanced ATF3 depletion induced ISC proliferation. However, heterozygous mutations cannot rescue the ATF3 depletion induced susceptibility to infection. In addition, *puc* overexpression cannot rescue ISC proliferation upon ATF3 loss. These results suggest that the control of JNK activity in the intestine is complex and ATF3-Raw controls JNK pathway independent of Puckered. We have rephrased the conclusion sentence in the results part of the revised manuscript (Line 20-21, page 12).

“2) I would suggest the authors remove the repression of ATF3 by itself, presented in Fig 7D “Predicted model for ATF3-raw regulates JNK activity”. They do have data showing ATF3 binds to its own genomic locus, but this does not tell if those binding sites are required for activation or repression.”

We provided evidence in the revised manuscript to show ATF3 binds to its own promoter. We also performed ChIP-qPCR to confirm the bound region on *ATF3* loci (Supplementary Fig. 3a,b). In addition, discussed in the revised manuscript, a previous study from the Uhlirova lab already showed ATF3 transcriptional level is dysregulated in *atf3* mutants (Rynes et al., 2012). Studies in mice also suggest that ATF3 transcriptionally represses itself expression (Wolfgang et al., 2000; Gargiulo et al., 2013). Taken together, we believe these data support a model whereby ATF3 represses its own expression. We have also added a sentence in the discussion that further mechanistic studies might be necessary.

Reviewer #3:

“The authors have addressed my concerns satisfactorily. However, there is a mistake in the labeling of the new panel K in figure 3. Raw over expression not Raw RNAi should suppress Puc LacZ. There was no experiment in which both ATF3 and raw RNAis are expressed in the corresponding panels g-j”

We thank the reviewer for pointing this out. We have corrected the labeling in Fig. 3k in the revised manuscript.

Reviewer #4:

“The authors have responded appropriately to my original comments, performing additional experiments that point to a role for p38 MAPKinase in the control of ATF3 expression, and showing that Puc over-expression is not sufficient to inhibit ISC proliferation induced by the loss of ATF3.

The authors only include these findings as additional reviewer data in the rebuttal letter, but they add interesting additional insight into the regulation of JNK signaling by ATF3 and Raw, and should, for this reason, be included in the main manuscript as supplemental data.”

We thank the reviewer for his suggestion. We already added the experiment data regarding the mitotic index of *puc* overexpression in ATF3 loss flies in the Supplemental Fig.4h in the revised manuscript.

REFERENCES

Gargiulo, G. et al. In vivo RNAi screen for BMI1 targets identifies TGF- β /BMP-ER stress pathways as key regulators of neural- and malignant glioma-stem cell homeostasis. *Cancer Cell*. 23, 660-76 (2013).

Rynes, J. et al. Activating Transcription Factor 3 Regulates Immune and Metabolic Homeostasis. *Mol Cell Biol* 32, 3949-3962 (2012).

Wolfgang, C.D., Liang, G., Okamoto, Y., Allen, A.E. & Hai, T. Transcriptional autorepression of the stress-inducible gene ATF3. *J Biol Chem*. 275, 16865-70 (2000).